# From Reward-Free Representations to Preferences:
# Rethinking Offline Preference-Based Reinforcement Learning

**Jun-Jie Yang** [1]   **Chia-Heng Hsu** [1]   **Kui-Yuan Chen** [1]   **Ping-Chun Hsieh** [1]

## Abstract

Preference-based reinforcement learning (PbRL) avoids explicit reward engineering by learning from pairwise human preference feedback. Existing offline PbRL methods typically follow a two-stage pipeline, first learning a reward or preference model from labeled preferences and then performing offline RL on unlabeled data. We revisit offline PbRL through the lens of reward-free representation learning (RFRL) from the zero-shot RL literature, and propose a new training framework that first learns latent successor-measure representations from reward-free offline data, followed by contrastive search and fine-tuning using preference data. Through extensive experiments and ablations, we show that our method achieves superior preference efficiency over offline PbRL baselines. This work is the first to connect RFRL with PbRL, highlighting its potential as a feedback-efficient solution. Our code is publicly available at `https://github.com/rl-bandits-lab/FB-PbRL`.

## 1. Introduction

Reinforcement learning (RL) has achieved remarkable success in sequential decision-making problems where reward signals can be well specified, including game playing (Silver et al., 2016; Vinyals et al., 2019), recommender systems (Zheng et al., 2018; Afsar et al., 2022), and resource scheduling and optimization (Khalil et al., 2017; Zhang et al., 2020). Despite these advances, a long-standing practical bottleneck in RL lies in reward design: crafting reward functions that faithfully encode task objectives is often difficult, ambiguous, or even infeasible in complex real-world settings (Skalse et al., 2022). To address this challenge, preference-based reinforcement learning (PbRL) has emerged as a compelling alternative to explicit reward engineering (Christiano et al., 2017; Ouyang et al., 2022). Instead of relying on manually specified rewards, PbRL learns from trajectories augmented with pairwise preference feedback provided by a human annotator. By leveraging relative comparisons rather than scalar rewards, PbRL provides a flexible mechanism for aligning learned policies with human intent while avoiding brittle reward design.

Most existing PbRL methods adopt a two-stage training paradigm. In the first stage, a reward or preference model is learned from human feedback via supervised learning; in the second stage, this learned model serves as a surrogate objective for downstream RL, typically over large unlabeled offline datasets, as illustrated in Figures 1a and 1b. However, human preference feedback is expensive to obtain, and practical PbRL systems must operate under severe labeling constraints. With limited preference data, reward models are prone to overfitting and poor generalization (An et al., 2023; Ye et al., 2024; Bai et al., 2025), while directly learning a preference model often leads to underfitting and low predictive accuracy. Enabling effective PbRL under scarce feedback therefore remains a critical challenge.

In parallel, a growing line of work has investigated *reward-free representation learning* (RFRL) to improve sample efficiency in reward-based RL (Touati & Ollivier, 2021; Touati et al., 2023; Agarwal et al., 2025; Chen et al., 2026). RFRL aims to learn compact, general-purpose representations of states or state–action pairs purely from reward-free offline data. Typically, RFRL proceeds in two stages: (1) *reward-free pre-training*, where representations are learned by modeling long-horizon environment dynamics without task-specific rewards; and (2) *test-time reward-based task specification*, where these representations enable sample-efficient learning or even zero-shot generalization once explicit rewards are provided (Pirotta et al., 2024; Tirinzoni et al., 2025).

Despite the conceptual alignment between PbRL and RFRL, both of which seek to learn effective behavior under weak or absent reward supervision, their connection has not yet been explored. In PbRL, tasks are specified through trajectory-level preference feedback, which constitutes a significantly weaker signal than per-step rewards. A naive attempt to bridge the two paradigms would be to first fit a reward

---

*Equal contribution  [1] Department of Computer Science, National Yang Ming Chiao Tung University, Hsinchu, Taiwan . Correspondence to: Ping-Chun Hsieh <pinghsieh@nycu.edu.tw>.

*Proceedings of the $43^{rd}$ International Conference on Machine Learning*, Seoul, South Korea. PMLR 306, 2026. Copyright 2026 by the author(s).

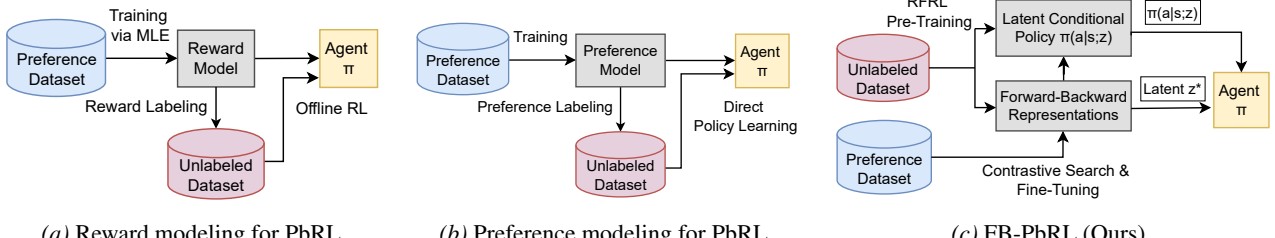

*(a)* Reward modeling for PbRL.  *(b)* Preference modeling for PbRL.  *(c)* FB-PbRL (Ours).

*Figure 1.* An illustration of the commonly used PbRL training pipelines and our proposed FB-PbRL framework.

model from preferences (*e.g.,* under the Bradley–Terry formulation) and then use this reward for RFRL-style task specification. However, this approach inherits the same reward over-optimization and generalization issues in conventional PbRL, particularly when preference data is scarce. This motivates a fundamental and yet underexplored question: *Can RFRL inform PbRL?*

In this work, we answer this question in the affirmative. We show that representations learned through RFRL pre-training can serve as compact and informative features for preference learning, fundamentally reshaping the PbRL pipeline. Our key insight is that, under mild structural conditions, the standard preference loss in PbRL can be equivalently reformulated as a contrastive objective akin to SimCLR (Chen et al., 2020) when applied to RFRL representations. Building on this insight, we propose a new two-stage offline PbRL framework: (1) *RFRL pre-training*, where we adopt the Forward–Backward (FB) decomposition method (Touati & Ollivier, 2021) to learn latent state–action representations and a latent-conditioned policy from reward-free offline data; and (2) *preference-guided search and fine-tuning*, where we apply a contrastive objective over preference-labeled trajectory segments to jointly fine-tune the FB representations and search for the latent vector required by the policy. This eliminates the need to explicitly learn either a reward or preference model. An overview of the proposed framework is shown in Figure 1c.

We summarize our main contributions as follows:

- **A new perspective on PbRL through RFRL**: We uncover a fundamental connection between PbRL and RFRL, showing that PbRL objectives can be interpreted through the lens of contrastive representation learning.
- **A new training pipeline for PbRL**: We introduce FB-PbRL, a two-stage framework that combines RFRL pre-training with SimCLR-style contrastive fine-tuning, obviating the need to learn a reward or preference model.

- **Empirical insights into RFRL for PbRL**: We evaluate FB-PbRL on a diverse set of offline PbRL benchmarks, including locomotion and navigation tasks from the Deep-Mind Control Suite and robotic manipulation tasks from Adroit and MetaWorld. Our results show that FB-PbRL consistently outperforms state-of-the-art PbRL methods,

achieves competitive or superior performance with substantially fewer preference labels, and remains robust under noisy preference feedback. We further observe favorable fine-tuning efficiency in wall-clock time.

## 2. Background

In this section, we formally describe the standard offline PbRL and RFRL settings and set up the notation required throughout the paper. We consider the RL problem formulated by a Markov decision process (MDP) characterized by the tuple $(\mathcal{S}, \mathcal{A}, p_0, \mathcal{T}, r, \gamma)$, where $\mathcal{S}$ and $\mathcal{A}$ denote the state and action spaces, $p_0 \in \Delta(\mathcal{S})$[1] is the initial state distribution, $\mathcal{T} : \mathcal{S} \times \mathcal{A} \to \Delta(\mathcal{S})$ is the transition function, $r : \mathcal{S} \times \mathcal{A} \to \mathbb{R}$ is the scalar reward function, and $\gamma \in [0, 1)$ is the discount factor. Let $\Pi := \{\pi | \pi : \mathcal{S} \to \Delta(\mathcal{A})\}$ denote the set of all Markovian policies. The objective is to learn a policy $\pi \in \Pi$ that maximizes the expected total discounted reward $\mathcal{J}(\pi) = \mathbb{E}_{s_0 \sim p_0, \pi} \left[ \sum_{t=0}^{\infty} \gamma^t r(s_t, a_t) \right]$, where $s_t$ and $a_t$ denote the observed state and action taken at time $t$.

### 2.1. Offline PbRL

We consider the standard offline PbRL setting as follows: (i) The agent learns solely from a static dataset of unlabeled, reward-free trajectories $\mathcal{D}$ pre-collected by one or multiple behavior policies. No further online interaction with the environment is allowed to the agent. (ii) Regarding supervision signals, the agent is given a dataset of pairwise human preference feedback on some of the trajectory segments in $\mathcal{D}$. Specifically, let $\sigma = ((s_0, a_0), \ldots, (s_{k-1}, a_{k-1}))$ denote a segment of fixed length $k$ sampled from $\mathcal{D}$, and the feedback label $y \in \{1, -1, 0\}$ denotes a preference for the first segment ($\sigma^{(1)} \succ \sigma^{(2)}$), the second segment ($\sigma^{(2)} \succ \sigma^{(1)}$), or an equal preference, respectively. Each feedback is stored in the preference dataset $\mathcal{D}_{\text{pref}}$ as a triple $(\sigma^{(1)}, \sigma^{(2)}, y)$.

In PbRL, the Bradley-Terry (BT) model (Bradley & Terry, 1952) is commonly used to formulate the connection between preferences and the rewards, assuming that the probability of preferring $\sigma^{(1)}$ over $\sigma^{(2)}$ depends on the segment-

---

[1] We let $\Delta(\mathcal{X})$ denote the set of all probability distributions over a measurable set $\mathcal{X}$ throughout this paper.

level cumulative rewards:

$$P_{\psi}(\sigma^{(1)} \succ \sigma^{(2)}) = \frac{\exp\left(\sum_{t=0}^{k-1} r_{\psi}(s_t^{(1)}, a_t^{(1)})/\tau\right)}{\sum_{i \in \{1,2\}} \exp\left(\sum_{t=0}^{k-1} r_{\psi}(s_t^{(i)}, a_t^{(i)})/\tau\right)},$$

(1)

where $r_{\psi}$ is a parametrized reward function that assigns scores to state-action pairs and $\tau > 0$ is the temperature parameter. As a result, $r_{\psi}$ can be optimized through minimizing the cross-entropy preference loss:

$$\mathcal{L}(\psi) := -\mathbb{E}_{\mathcal{D}_{\text{pref}}}\Big[\mathbb{I}(y = 1) \log P_{\psi}(\sigma^{(1)} \succ \sigma^{(2)})$$
$$+ \mathbb{I}(y = -1) \log P_{\psi}(\sigma^{(2)} \succ \sigma^{(1)})\Big],$$

(2)

where $\mathbb{I}(\cdot)$ is the indicator function. Once the reward parameters $\psi$ are learned, one can use the reward model to label the transitions in the offline dataset $\mathcal{D}$ and employ an off-the-shelf offline RL algorithm for policy learning.

### 2.2. RFRL and Forward-Backward Representations

In the typical RFRL setting, the goal is to learn compact and general-purpose representations that can be used for any downstream tasks, primarily via unsupervised pre-training from reward-free offline data $\mathcal{D}$. Among the RFRL methods, one pivotal machinery is the *successor measure* defined as

$$\mathcal{M}^{\pi}(s, a, X) := \mathbb{E}_{\pi}\Big[\sum_{t=0}^{\infty} \gamma^t \mathbb{I}((s_t, a_t) \in X)\Big|s_0 = s, a_0 = a\Big],$$

(3)

which essentially captures the discounted future occupancy of the state-action pairs in $X \subseteq \mathcal{S} \times \mathcal{A}$ if the agent starts from $(s, a)$ and follows a policy $\pi$. Define the standard Q function under a policy $\pi$ with respect to a reward function $r$ as $Q_r^{\pi}(s, a) := \mathbb{E}_{\pi}[\sum_{t=0}^{\infty} \gamma^t r(s_t, a_t) \mid s_0 = s, a_0 = a]$. Notably, one fundamental connection between the Q function with the successor measure is as follows[2]:

$$Q_r^{\pi}(s, a) = \sum_{(s', a') \in \mathcal{S} \times \mathcal{A}} \mathcal{M}^{\pi}(s, a, \{(s', a')\}) r(s', a'). \quad (4)$$

Consequently, for any reward function $r$, learning the successor measure of an optimal policy for $r$ (denoted by $\pi_r^*$) implies learning the optimal Q function $Q^{\pi_r^*}$.

To efficiently learn $\mathcal{M}^{\pi_r^*}$, the FB framework (Touati & Ollivier, 2021; Touati et al., 2023) decomposes the successor measure into the inner product of two encodings, *i.e.,*

$$\mathcal{M}^{\pi_r^*}(s, a, \{(s', a')\}) = \mathbf{F}_{\theta}(s, a, \boldsymbol{z}_r)^{\top} \mathbf{B}_{\omega}(s', a'), \quad (5)$$

where $\mathbf{F}_{\theta} : \mathcal{S} \times \mathcal{A} \times \mathbb{R}^d \to \mathbb{R}^d$ and $\mathbf{B}_{\omega} : \mathcal{S} \times \mathcal{A} \to \mathbb{R}^d$ are the

---

[2]For ease of exposition, here we presume countable state and action spaces. The same argument can be directly extended to the uncountable case by a proper measure-theoretic argument.

*Forward representation* and the *Backward representation*, respectively, and $\boldsymbol{z}_r \in \mathbb{R}^d$ is a latent task specification vector to be designed to encode reward information. By combining (4)-(5), the FB method designs $\boldsymbol{z}_r$ such that the optimal Q function for reward function $r$ can be expressed as

$$Q^{\pi_r^*}(s, a) = \mathbf{F}_{\theta}(s, a, \boldsymbol{z}_r)^{\top} \boldsymbol{z}_r, \quad (6)$$

where the task specification vector $\boldsymbol{z}_r$ is designed to capture reward information along with Backward representation as

$$\boldsymbol{z}_r = \mathbb{E}_{(s,a) \sim \mathcal{D}}\big[\mathbf{B}_{\omega}(s, a) \, r(s, a)\big]. \quad (7)$$

Through this design, if $\mathbf{F}_{\theta}$ and $\mathbf{B}_{\omega}$ are well learned via pre-training, an optimal policy for a reward function $r$ can be directly recovered in a zero-shot manner at test time, *i.e.,*

$$\pi_r^*(s) = \arg\max_{a \in \mathcal{A}} \mathbf{F}_{\theta}(s, a, \boldsymbol{z}_r)^{\top} \boldsymbol{z}_r. \quad (8)$$

## 3. Methodology

In this section, we formally describe the proposed FB-PbRL framework. We identify two key limitations of naively applying reward-based and preference-based task specification to FB representations and propose two corresponding strategies integrated within FB-PbRL.

### 3.1. Test-Time Preference-based Task Specification

The standard FB framework enables zero-shot policy inference but relies on scalar reward signals $r(s, a)$ to construct the latent task specification vector $\boldsymbol{z}_r$. This reliance is problematic in offline PbRL, where supervision is provided only through pairwise preferences over trajectory segments rather than explicit rewards.

**Naive task specification via reward modeling.** A straightforward way to adapt FB-based RFRL to PbRL is to first learn a proxy reward function $r_{\psi}(s, a)$ from preference data (e.g., via Equation (2)) and then use it for task specification at test time. However, similar to standard PbRL, this approach is vulnerable to reward over-optimization. As shown in Figure 2, the rewards learned under BT modeling exhibit a large mismatch with the ground-truth rewards, collapsing toward the mid-range and failing to capture the underlying true reward structure. Consequently, reward-based task specification yields substantially lower evaluation returns compared to task specification using true rewards.

**Key idea: Connecting RFRL with SimCLR contrastive learning.** Alternatively, we propose to exploit the intrinsic structure of the FB latent space and directly infer the task specification vector without explicitly learning a reward model. Since the FB latent space is designed to span all reward functions (Touati & Ollivier, 2021), we can interpret the task specification problem as finding a direction $\boldsymbol{z} \in \mathbb{R}^d$ that best aligns with the observed preferences. Our key

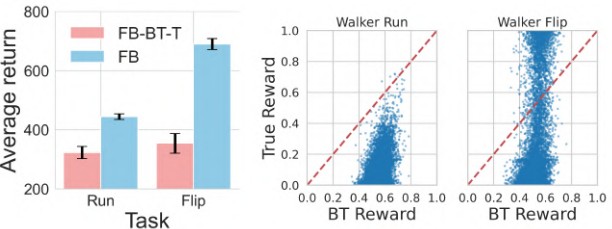

*Figure 2.* **Reward over-optimization under the proxy rewards on *Walker-run* and *Walker-flip*.** "FB-BT-T" performs test-time task inference using a proxy BT-based reward function, whereas "FB" uses ground-truth rewards in task specification. (a) The bar chart reports average return with error bars indicating standard deviation over 5 seeds. (b) The scatter plot visualizes predicted rewards under BT modeling against the ground-truth rewards.

insight is that, under mild structural conditions, the standard preference loss in PbRL can be equivalently reformulated as a contrastive objective akin to SimCLR (Chen et al., 2020).

Let $\mathbf{F}_{\bar{\theta}}$ and $\mathbf{B}_{\bar{\omega}}$ denote the pre-trained Forward and Backward representations, respectively. Consider a class of linearly parametrized reward functions with respect to the pre-trained Backward representations, *i.e.,*

$$r_{\psi}(s,a) := \mathbf{B}_{\bar{\omega}}(s,a)^{\top}\psi, \qquad (9)$$

where $\psi \in \mathbb{R}^d$. Under linear realizability, we can derive the corresponding latent vector $z$ for each reward function as

$$z_{\psi} := \mathbb{E}_{(s,a)\sim\mathcal{D}}\big[\mathbf{B}_{\bar{\omega}}(s,a)r_{\psi}(s,a)\big] \qquad (10)$$

$$= \underbrace{\Big(\mathbb{E}_{(s,a)\sim\mathcal{D}}\big[\mathbf{B}_{\bar{\omega}}(s,a)\mathbf{B}_{\bar{\omega}}(s,a)^{\top}\big]\Big)}_{=:\mathbf{H}_{\mathbf{B}}}\psi, \qquad (11)$$

where $\mathbf{H}_{\mathbf{B}}$ is a $d \times d$ and non-negative definite matrix. Clearly, $\mathbf{H}_{\mathbf{B}}$ would remain fixed if $\mathbf{B}_{\bar{\omega}}$ is fixed after pre-training. The above suggests that if $\mathbf{H}_{\mathbf{B}}$ is invertible, then there exists a bijection between $z_{\psi}$ and $\psi$, *i.e.,* $\psi = \mathbf{H}_{\mathbf{B}}^{-1}z_{\psi}$. For notational convenience, for any trajectory segment $\sigma = \big((s_1,a_1),\ldots,(s_k,a_k)\big)$, we define

$$\mathbf{B}_{\bar{\omega}}(\sigma) := \frac{1}{k}\sum_{i=1}^{k}\mathbf{B}_{\bar{\omega}}(s_i,a_i). \qquad (12)$$

Accordingly, by using (9)-(11), we can rewrite the preference loss in (2) in terms of latent $z$ as

$$\mathcal{L}_{\text{pref}}(z;\bar{\omega}) = -\mathbb{E}\Big[\log\frac{\exp(z^{\top}\mathbf{H}_{\mathbf{B}}^{-1}z_{\sigma}^{+})}{\exp(z^{\top}\mathbf{H}_{\mathbf{B}}^{-1}z_{\sigma}^{+}) + \exp(z^{\top}\mathbf{H}_{\mathbf{B}}^{-1}z_{\sigma}^{-})}\Big], \qquad (13)$$

where the expectation is taken over some sampling distribution of $k$-step preference pairs $(\sigma^+, \sigma^-)$, the temperature $\tau = k$, and the two latent vectors are defined as

$$z_{\sigma}^{+} := \mathbf{B}_{\bar{\omega}}(\sigma^{+}), \; z_{\sigma}^{-} := \mathbf{B}_{\bar{\omega}}(\sigma^{-}). \qquad (14)$$

Notably, the equivalence to the SimCLR objective precisely holds under two structural conditions that are standard within the FB framework: (i) *Linear reward parameterization* as defined in (9); and (ii) *Orthonormality of the backward embedding* ($\mathbf{H}_{\mathbf{B}} = \mathbf{I}_d$), which approximately holds in practice since standard FB pre-training imposes orthonormality on $\mathbf{B}_{\bar{\omega}}$ through an auxiliary loss (Touati & Ollivier, 2021). These conditions make explicit that the equivalence follows directly from the latent linear reward structure, thereby providing a principled bridge between FB representations and preference optimization. A detailed derivation of this connection is provided in Appendix B.1.

**Technique 1: Contrastive preference-based task specification (CPTS).** As suggested by (13), a proper latent task specification (denoted by $z_{\text{CPTS}}^*$) can be derived at test time by minimizing the contrastive preference loss in (13), *i.e.,*

$$z_{\text{CPTS}}^* := \arg\min_{z}\mathcal{L}_{\text{pref}}(z;\bar{\omega}). \qquad (15)$$

Intuitively, the SimCLR loss encourages $z$ to align with the latent embeddings associated with preferred trajectory segments while repelling it from non-preferred ones.

Notably, the proposed CP-TS technique enjoys two nice features: (i) *No reward model learning*: The latent task specification can be determined directly based on the preference dataset $\mathcal{D}_{\text{pref}}$, without explicitly learning a reward model. (ii) *Optimization of a convex function with respect to a low-dimensional decision variable*: Crucially, as $\mathcal{L}_{\text{pref}}$ in (13) in convex in $z$ and $z$ is usually a low-dimensional vector in practice (*e.g.,* a few hundred dimensions), the resulting optimization with respect to $z$ is significantly simpler than learning an explicit preference or reward model, which typically requires training a high-capacity neural network.

Specifically, even if cosine similarity is used, it reduces to an inner product, $\cos(z, z_{\sigma}) = z^{\top}z_{\sigma}$, because $z$ and $z_{\sigma}$ are normalized by default in the FB framework, thereby retaining the convex structure. While jointly optimizing the representation network is inherently non-convex, our convexity guarantee specifically applies to this test-time search where the representation is frozen and we solve solely for $z$.

Crucially, this test-time search procedure comes with formal theoretical guarantees. By extending the analysis of (Touati & Ollivier, 2021) to the PbRL setting, we show that representation sufficiency reduces to preference data coverage and estimation error, yielding near-optimal downstream control. We provide the formal proposition and proof sketch in Section B.2.

### 3.2. Preference-Guided Fine-Tuning

**Limitations of purely test-time task specification.** Despite that test-time task specification is well-motivated, its effectiveness can be limited in practice due to the design of FB

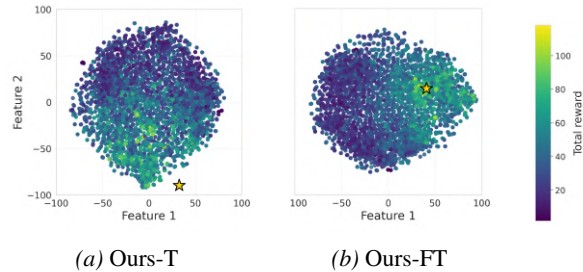

*(a)* Ours-T  *(b)* Ours-FT

*Figure 3.* **Reward-aligned latent geometry.** t-SNE visualization of the learned latent $z_\sigma \equiv \mathbf{B}_\omega(\sigma)$ on the *Walker-walk* task. **(a)** Test-time preference-based task specification without fine-tuning. **(b)** Preference-guided fine-tuning. The $z_\sigma$s of the offline trajectory segments are colored by ground-truth returns, with brighter hues indicating higher returns. The final task specifications $z^*_{\mathrm{CPTS}}$ and $z^*_{\mathrm{PGFT}}$ are marked by a star in (a) and (b), respectively.

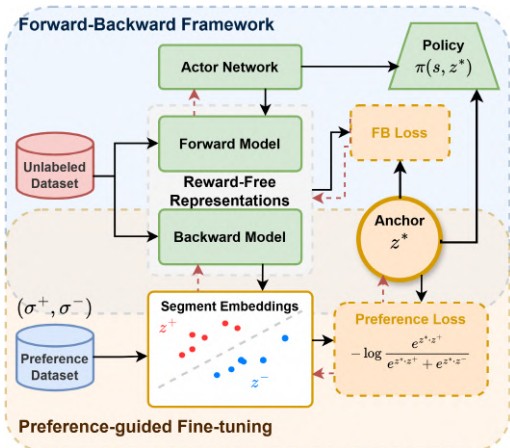

*Figure 4.* **The overall training pipeline of FB-PbRL.** The architecture comprises two integral parts: **(Top) Forward-Backward Framework:** Adopt FB decomposition to learn latent state-action representations and a latent-conditioned policy from offline data. **(Bottom) Preference-Guided Search and Fine-tuning:** Leverage segment embeddings and a contrastive objective over preference data to refine FB representations and optimize the anchor $z^*$, thereby conditioning the policy $\pi(s, z^*)$ for preference alignment.

pre-training. Specifically, as pre-training is meant to learn general-purpose representations across tasks and is unaware of the specific downstream task at test time, the $\mathbf{F}_{\bar{\theta}}$ and $\mathbf{B}_{\bar{\omega}}$ are typically pre-trained under task vectors $z$ sampled from a less informative prior, *e.g.,* standard normal $z \sim \mathcal{N}(0, I_d)$. As a result, the inferred $z^*_{\mathrm{CPTS}}$ under this pre-trained $\mathbf{B}_{\bar{\omega}}$ can likely be distant from those $(z^+_\sigma, z^-_\sigma)$ induced by the preference dataset as in (14). This phenomenon can be commonly observed in practice, *e.g.,* in Figure 3a.

**Technique 2: Preference-guided fine-tuning (PG-FT).** To tackle the above limitation, we propose to fine-tune the FB models jointly with the task specification (denoted by $z^*_{\mathrm{PGFT}}$). Instead of treating the FB's latent space as fixed, we adapt the $\mathbf{F}_\theta$ and $\mathbf{B}_\omega$ via fine-tuning guided by $\mathcal{L}_{\mathrm{pref}}$ in (13) conditioned on the current $z^*$ such that task-relevant directions are easier to identify and exploit.

The effect of PG-FT is illustrated in Figure 3. We have two empirical observations: (i) *Reward-aligned latent geometry*: PG-FT reshapes the latent geometry and produces a smoother reward landscape that effectively guides the anchor $z^*$ towards a well-performing task representation. (ii) *In-distribution latent $z^*$*: The final $z^*$ after fine-tuning is more likely to be "in-distribution" as $\mathbf{F}_\theta$ and $\mathbf{B}_\omega$ are fine-tuned along with $z^*$ under the guidance of $\mathcal{L}_{\mathrm{pref}}$.

### 3.3. FB-PbRL: Training Objectives and Implementation

In this subsection, we are ready to put everything together and summarize the full FB-PbRL algorithm. The detailed architecture is presented in Figure 4. Below we first describe the main loss components used in FB-PbRL.

**Measure Loss.** To learn the $\mathbf{F}_\theta$ and $\mathbf{B}_\omega$ networks, we leverage the standard measure loss in the FB framework (Touati et al., 2023), *i.e.,* minimizing the Bellman residual defined by $\mathbf{F}_\theta$ and $\mathbf{B}_\omega$ as follows

$$\mathcal{L}_{\mathrm{m}}(\theta, \omega; z) := \mathbb{E}_{\substack{(s,a,s') \sim \mathcal{D} \\ (s^\dagger, a^\dagger) \sim \mathcal{D}}} \Big[ \big(\mathbf{F}_\theta(s, a, z)^\top \mathbf{B}_\omega(s^\dagger, a^\dagger)$$
$$- \gamma\, \mathbf{F}_{\hat{\theta}}(s', \pi(s'), z)^\top\, \mathbf{B}_{\hat{\omega}}(s^\dagger, a^\dagger)\big)^2 \Big]$$
$$- 2\, \mathbb{E}_{(s,a,s') \sim \mathcal{D}} \Big[ \mathbf{F}_\theta(s, a, z)^\top \mathbf{B}_\omega(s', a') \Big]. \quad (16)$$

**Orthonormality Loss.** To preserve the required structural property of the Backward representation (*i.e.,* $\mathbf{H_B} \approx \mathbf{I}_d$), we impose an orthonormality regularization on $\mathbf{B}_\omega$ as in the standard FB framework, *i.e.,*

$$\mathcal{L}_{\mathrm{ortho}}(\omega) := \left\| \mathbb{E}_{(s,a) \sim \mathcal{D}} \big[ \mathbf{B}_\omega(s, a) \mathbf{B}_\omega(s, a)^\top \big] - \mathbf{I}_d \right\|_{\mathrm{F}}^2, \quad (17)$$

where $\|\cdot\|_{\mathrm{F}}$ denote the Frobenius norm. Due to the page limit, we defer the simplified version of (17) to Section A.

**Preference Loss.** As discussed in Sections 3.1 and 3.2, the contrastive preference loss $\mathcal{L}_{\mathrm{pref}}$ serves two roles: (i) under CPTS, it is minimized to search for the task specification $z$; and (ii) under PG-FT, its implicit dependence on $\mathbf{B}_\omega$ through $(z^+_\sigma, z^-_\sigma)$ enables fine-tuning of $\mathbf{B}_\omega$. Standard FB pre-training assumes latent vectors $z$ lie on a unit hypersphere, consistent with the normalization used in SimCLR-style contrastive learning. Let $\mathrm{sim}(\cdot, \cdot)$ denote cosine similarity. We define the preference loss as

$$\mathcal{L}_{\mathrm{pref}}(z, \omega) := -\mathbb{E}_{\mathcal{D}_{\mathrm{pref}}} \left[ \log \frac{e^{\mathrm{sim}(z, z^+_\sigma(\omega))}}{e^{\mathrm{sim}(z, z^+_\sigma(\omega))} + e^{\mathrm{sim}(z, z^-_\sigma(\omega))}} \right], \quad (18)$$

where the expectation is over sampled preference pairs $(\sigma^+, \sigma^-)$ from $\mathcal{D}_{\mathrm{pref}}$, and the dependence on $\mathbf{B}_\omega$ is through $z^+_\sigma(\omega)$ and $z^-_\sigma(\omega)$.

**Full FB-PbRL framework.** The proposed FB-PbRL method consists of two stages as follows.

**Algorithm 1** FB-PbRL

---

1: **Input:** Datasets $\mathcal{D}$ and $\mathcal{D}_{\text{pref}}$, pre-trained $\mathbf{F}_{\bar{\theta}}$ and $\mathbf{B}_{\bar{\omega}}$, loss coefficients $\lambda$ and $\alpha$, and initial $\boldsymbol{z}^* \in \mathbb{R}^d$.
2: **for** $t = 1, 2, \ldots$ **do**
3:     Sample a batch of transitions $\mathcal{B}_t$ from $\mathcal{D}$
4:     Update $\mathbf{F}_{\bar{\theta}}$ and $\mathbf{B}_{\bar{\omega}}$ by stochastic gradient on the loss $\mathcal{L}_{\text{m}}(\bar{\theta}, \bar{\omega}; \boldsymbol{z}^*) + \lambda \mathcal{L}_{\text{ortho}}(\bar{\omega})$ in (16)-(17) with $\mathcal{B}_t$
5:     Sample a batch of preferences $\mathcal{P}_t$ from $\mathcal{D}_{\text{pref}}$
6:     Compute $\mathcal{L}_{\text{pref}}(\boldsymbol{z}^*, \bar{\omega})$ as in (18) with $\mathcal{P}_t$
7:     Update $\mathbf{B}_{\bar{\omega}}$ and $\boldsymbol{z}^*$ by gradient on $\alpha \cdot \mathcal{L}_{\text{pref}}(\boldsymbol{z}^*, \bar{\omega})$
8:     Update policy $\pi$ based on $\mathbf{F}_{\bar{\theta}}$, $\mathbf{B}_{\bar{\omega}}$, and $\boldsymbol{z}^*$
9: **end for**

---

- **RFRL pre-training**: FB-PbRL adopts the measure loss in (16) and the orthonormality loss in (17) to pre-train the Forward and Backward representations.

- **Preference-guided search and fine-tuning**: In the fine-tuning stage, FB-PbRL alternates between two subroutines: (i) Search for a latent vector $\boldsymbol{z}^*$ based on the guidance of $\mathcal{L}_{\text{pref}}(\boldsymbol{z}; \omega)$, *e.g.,* via stochastic gradient descent. (ii) Conditioning on the current latent vector $\boldsymbol{z}^*$, fine-tune $\mathbf{F}_\theta$ and $\mathbf{B}_\omega$ based on $\mathcal{L}_{\text{m}}(\theta, \omega; \boldsymbol{z}^*)$, $\mathcal{L}_{\text{ortho}}(\omega)$, and $\mathcal{L}_{\text{pref}}(\omega; \boldsymbol{z}^*)$. One notable difference from FB pre-training is that $\boldsymbol{z}^*$ serves as a *dynamically evolving anchor* such that the $\mathbf{F}_\theta$ and $\mathbf{B}_\omega$ can specialize its latent geometry around the specific behavior required by the target task.

The FB-PbRL framework is illustrated in Figure 4, with pseudocode summarized in Algorithm 1 and a detailed version in Algorithm 2. Note that general FB supports both state- and state–action-dependent backward representations; following the original implementation (Touati et al., 2023), we adopt state-dependent representations in the experiments.

## 4. Experiments

### 4.1. Experimental Setup

**Tasks and Datasets.** To avoid the survival instinct issue (Li et al., 2023) in D4RL (Fu et al., 2020), we primarily evaluate on the **DeepMind Control Suite (DMC)** (Tassa et al., 2018). We consider 16 tasks across four domains: three locomotion domains (*Cheetah*, *Walker*, *Quadruped*) and the *Pointmass* goal-reaching domain. Following standard zero-shot RL practice, we use unsupervised RND-based datasets from ExORL (Yarats et al., 2022), representing a challenging low-quality regime without reward supervision. We further evaluate generalization on human preference datasets for robotic manipulation, including (i) the **Adroit** *Pen* dataset from Preference Transformer (Kim et al., 2023) and (ii) the **MetaWorld** (Yu et al., 2020) *Button-Press-Topdown* dataset from LiRE (Choi et al., 2024).

For DMC tasks, we use a scripted teacher (Lee et al., 2021a;

Mu et al., 2025) to generate synthetic preferences from ground-truth returns, with a tie condition to model human ambiguity when return differences are small (see Appendix E.2). For Pointmass, preferences are deterministic to avoid excessive ambiguity. Performance is measured by episodic return for DMC and Adroit, and success rate for MetaWorld. Additional details are in Appendix D.

**Offline PbRL Baselines.** We compare FB-PbRL with representative offline PbRL methods spanning diverse design choices. **DPPO** (An et al., 2023) and **OPPO** (Kang et al., 2023) incorporate contrastive learning into PbRL, while **OPRL** (Shin et al., 2023) and the state-of-the-art **CLAR-IFY** (Mu et al., 2025) use active query selection to improve label efficiency. These methods are allowed to actively query additional feedback from the offline dataset, consistent with their intended use. We also include **LiRE** (Choi et al., 2024), which leverages listwise reward estimation for richer supervision.

**RFRL and Zero-Shot RL Baselines.** We additionally compare against representative RFRL methods that learn reward-free representations for zero-shot generalization. This includes **FB** (Touati & Ollivier, 2021), which forms the backbone of our framework, as well as geometry- and structure-aware methods such as **Laplacian** (Wu et al., 2019) and **HILP** (Park et al., 2024). We further evaluate recent approaches including **PSM** (Agarwal et al., 2025) and **RLDP** (Jajoo et al., 2025). All RFRL baselines require access to ground-truth rewards at test time for task specification, whereas FB-PbRL operates solely on preference data. We re-implement FB for consistency, and report results for other zero-shot methods from (Jajoo et al., 2025). Implementation details are in Appendix E.2.

**Experiment Protocols.** We follow standard PbRL benchmarks using a fixed budget of 2,000 preference pairs (termed **PbRL Protocol**). To align with zero-shot RL baselines that access 10,000 reward-labeled transitions at test time, we also adopt this setup (termed **Zero-Shot RL Protocol**), where preferences are constructed from a subset of 400 trajectory segments (10,000 transitions), ensuring comparable supervision scale to (Jajoo et al., 2025). Unless stated otherwise, all the results are reported as mean ± std over 5 seeds.

### 4.2. Main Results

**Comparison with offline PbRL baselines.** Table 1 compares FB-PbRL with offline PbRL baselines on DMC tasks, where low-quality offline datasets severely limit existing methods and prevent effective policy learning. In contrast, FB-PbRL outperforms nearly all baselines even with contrastive learning applied only at test time, indicating that its representations remain discriminative under severe distribution shifts. Full fine-tuning further improves performance, achieving the best results across tasks, with Ours-FT exceeding a score of 900 on over half of them. These results

*Table 1.* **Performance on DMC under the PbRL Protocol.** *Ours-T* denotes FB-PbRL with test-time CPTS only, while *Ours-FT* denotes the full FB-PbRL framework with fine-tuning. The yellow and gray shading indicate the best and second-best performances of each row.

| Domain | Task | DPPO | OPPO | OPRL | CLARIFY | LIRE | Ours-T | Ours-FT |
|---|---|---|---|---|---|---|---|---|
| Cheetah | Run | 71.2±19.8 | 97.9±14.5 | 115.1±8.7 | 106.7±5.1 | 109.1±14.0 | 207.6±29.9 | 249.5±74.5 |
| | Run Backward | 74.4±17.9 | 113.1±8.0 | 99.9±7.2 | 93.6±11.0 | 135.6±28.1 | 160.7±44.2 | 333.9±28.5 |
| | Walk | 285.1±52.6 | 258.4±26.4 | 445.1±9.3 | 446.5±32.5 | 419.3±23.8 | 665.9±91.9 | 920.3±30.7 |
| | Walk Backward | 378.2±73.2 | 533.2±54.5 | 445.5±34.9 | 439.3±36.8 | 589.7±170.3 | 660.4±130.9 | 983.3±0.5 |
| | *Average* | 202.3 | 200.9 | 276.4 | 271.5 | 313.4 | 344.7 | 621.7 |
| Walker | Walk | 218.0±9.5 | 219.6±3.7 | 214.9±2.6 | 214.3±4.2 | 199.1±2.8 | 714.3±138.5 | 961.5±4.2 |
| | Stand | 403.1±11.3 | 413.4±6.9 | 436.5±6.5 | 423.1±6.0 | 394.5±1.8 | 677.7±38.2 | 980.3±3.4 |
| | Run | 91.8±3.1 | 93.6±0.9 | 96.4±2.1 | 94.8±1.9 | 88.8±0.6 | 282.8±28.0 | 503.7±14.4 |
| | Flip | 256.4±11.3 | 263.2±4.8 | 267.4±5.7 | 263.3±3.4 | 247.5±1.7 | 458.7±30.4 | 606.0±7.9 |
| | *Average* | 242.3 | 247.5 | 253.8 | 248.9 | 232.5 | 533.4 | 762.9 |
| Quadruped | Run | 204.5±39.2 | 395.5±9.4 | 457.8±7.2 | 444.7±8.7 | 246.8±20.3 | 503.4±30.1 | 589.9±36.3 |
| | Walk | 194.4±73.7 | 398.7±33.3 | 420.6±32.5 | 350.6±15.9 | 284.3±43.7 | 606.1±35.2 | 944.2±8.9 |
| | Stand | 458.2±57.9 | 779.9±29.1 | 917.9±21.7 | 901.0±17.6 | 523.7±58.6 | 886.1±28.3 | 991.1±0.3 |
| | Jump | 379.1±104.3 | 703.0±19.0 | 728.1±2.2 | 715.4±12.9 | 459.9±89.0 | 658.1±42.6 | 862.2±19.8 |
| | *Average* | 309.1 | 569.3 | 631.1 | 602.9 | 378.7 | 663.4 | 846.9 |
| PointMass | Top Left | 52.4±37.4 | 59.8±9.2 | 558.1±71.9 | 536.6±68.0 | 160.7±21.2 | 70.2±76.3 | 926.6±30.8 |
| | Top Right | 4.1±1.0 | 19.4±3.5 | 399.7±125.7 | 287.2±41.5 | 106.1±33.4 | 77.8±76.3 | 597.8±148.5 |
| | Bottom Left | 3.9±1.2 | 11.9±3.8 | 338.8±29.2 | 369.0±131.7 | 95.2±16.0 | 116.7±120.2 | 752.8±67.0 |
| | Bottom Right | 4.8±1.6 | 5.5±1.2 | 53.2±35.8 | 78.2±29.9 | 47.2±12.8 | 11.7±20.2 | 6.2±6.1 |
| | *Average* | 16.3 | 24.1 | 337.5 | 317.8 | 102.3 | 69.1 | 570.8 |

demonstrate that FB-PbRL effectively leverages preference supervision to overcome prior limitations.

**Comparison with zero-shot RFRL baselines.** Table 2 compares FB-PbRL, which relies solely on preferences, with RFRL baselines that access ground-truth rewards at test time. Despite lacking reward supervision, Ours-FT achieves the strongest overall performance, outperforming most baselines across tasks and exceeding them by over 200 returns on average in the *quadruped* domain. These results show that preference-guided fine-tuning effectively reshapes the latent space, allowing FB-PbRL to adapt to task objectives without external rewards.

Despite that FB-PbRL achieves strong overall performance, we observe higher variance and lower returns on some Point-Mass tasks, particularly the *Bottom Right* goal in Table 1. This behavior is mainly due to coverage imbalance in the RND PointMass dataset and limited preference information from 10k transitions with short trajectory segments, which together provide weaker supervision in sparsely visited regions. Supporting visitation heatmaps and trajectory visualizations are provided in Appendix F.8.

**Robustness to limited and noisy preferences.** (1) **Impact of limited feedback**: A key concern in PbRL is whether effective supervision can be obtained from few preferences. Figure 5a shows that FB-PbRL degrades gracefully as the number of preference pairs decreases. Reducing the budget from 2,000 to 200 pairs results in only about 10% performance drop. Across all budgets, FB-PbRL consistently outperforms PSM, the strongest RFRL baseline in Table 2, and OPRL, the best offline PbRL baseline in most domains, demonstrating robustness under limited feedback. (2) **Robustness to noisy preferences**: To model imperfect human

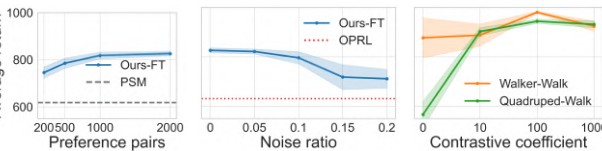

*(a)* Limited feedback *(b)* Noisy preference *(c)* Sensitivity to $\alpha$

*Figure 5.* **Robustness analysis of FB-PbRL. (a)** Performance scaling with the number of preference pairs under the **zero-shot RFRL protocol**; the dashed line denotes the strongest zero-shot RFRL baseline on *Quadruped*. **(b)** Robustness to noisy preference annotations under the **PbRL protocol**, with the dashed line indicating the best offline PbRL baseline. Results in (a) and (b) are averaged over four *Quadruped* tasks. **(c)** Sensitivity to the preference coefficient on *Walker-Walk* and *Quadruped-Walk*. Shaded regions denote standard deviation over 5 seeds.

feedback, we introduce a noise ratio $\delta$ that flips preferences with probability $\delta$ (Lee et al., 2021a). Figure 5b shows FB-PbRL exhibits only a gradual performance decline as noise increases and continues to outperform all baselines even at $\delta = 0.2$, indicating robustness under realistic supervision.

**Evaluation on robot arm manipulation with real human preferences.** To complement scripted-teacher experiments, we evaluate FB-PbRL on real human-labeled preference datasets, including Adroit *Pen* and MetaWorld *Button-Press-Topdown*, which involve high-dimensional observations and sparse success signals. We compare against DPPO and LiRE, the only offline PbRL methods applicable to static datasets, excluding active-querying methods such as OPRL and CLARIFY. DPPO results on Adroit Pen are taken from the original paper. As shown in Table 3, FB-PbRL performs strongly across tasks. While DPPO slightly outperforms FB-PbRL on *Pen-human*, our method remains competitive, with

*Table 2.* **Performance on DMC under the Zero-Shot RFRL Protocol.** *Ours-FT* denotes the full FB-PbRL framework with fine-tuning. The yellow and gray shading indicate the best and second-best performances of each row. Baseline methods sample 10,000 transitions with ground-truth reward labels at test time, whereas Ours uses 2,000 preference pairs constructed from 10,000 transitions.

| Domain | Task | Laplace | FB | HILP | PSM | RLDP | Ours-FT |
|---|---|---|---|---|---|---|---|
| | | | | *Ground-truth reward* | | | *Preferences* |
| Cheetah | Run | 96.3±35.7 | 169.1±15.1 | 68.2±47.1 | **244.4±80.0** | 236.3±20.8 | **338.8±47.1** |
| | Run Backward | 106.4±29.4 | 153.3±34.1 | 38.0±25.2 | 296.4±20.1 | **322.1±39.3** | **335.4±16.1** |
| | Walk | 409.2±56.1 | 509.7±59.7 | 318.3±168.4 | **984.2±0.5** | 895.3±49.8 | 924.6±72.6 |
| | Walk Backward | 654.3±219.8 | 710.5±140.1 | 349.6±236.3 | 979.0±7.7 | **984.8±0.9** | 982.8±1.0 |
| | *Average* | 316.5 | 385.6 | 193.5 | 626.0 | 609.6 | 645.4 |
| Walker | Walk | 190.5±168.5 | **823.6±11.3** | 399.7±39.3 | **891.4±46.8** | 790.9±67.6 | 787.0±37.5 |
| | Stand | 243.7±151.4 | **922.1±6.8** | 607.1±165.3 | 872.6±38.8 | 877.7±45.0 | **983.2±7.5** |
| | Run | 63.7±31.0 | **444.1±9.4** | 107.8±34.2 | 351.5±19.5 | 324.9±54.6 | **464.6±33.0** |
| | Flip | 48.7±17.7 | **689.7±18.4** | 278.0±59.6 | 640.8±31.9 | 492.9±22.8 | 562.7±22.3 |
| | *Average* | 136.7 | 719.9 | 348.1 | 689.1 | 621.6 | 699.4 |
| Quadruped | Run | 413.0±54.0 | 397.4±5.8 | 205.4±47.9 | 431.8±44.7 | **457.4±74.7** | **575.0±18.8** |
| | Walk | 494.6±62.5 | 529.8±45.7 | 218.5±86.7 | **604.0±73.7** | 465.4±185.3 | **889.4±31.7** |
| | Stand | 854.5±41.5 | 720.2±14.7 | 409.5±97.6 | **842.9±82.2** | 794.9±43.3 | **990.1±0.5** |
| | Jump | 642.8±114.2 | 599.1±15.4 | 325.5±93.1 | 596.4±94.2 | **733.3±55.3** | **850.9±14.5** |
| | *Average* | 601.2 | 561.7 | 289.8 | 618.7 | 612.8 | 826.3 |
| PointMass | Top Left | 713.5±58.9 | 489.7±155.7 | **944.5±12.9** | 831.4±69.5 | 890.4±60.8 | **928.1±22.4** |
| | Top Right | 581.1±214.8 | **761.2±30.2** | 96.0±166.3 | 730.3±58.1 | **795.6±21.1** | 483.5±174.2 |
| | Bottom Left | **689.1±37.1** | 107.2±83.2 | 192.3±177.5 | 451.4±73.5 | **805.2±20.4** | 541.5±268.0 |
| | Bottom Right | 21.3±42.5 | **236.1±171.9** | 0.17±0.3 | 43.3±38.4 | **193.4±167.6** | 51.4±60.4 |
| | *Average* | 501.2 | 398.5 | 308.3 | 514.1 | 671.1 | 501.1 |

the gap likely due to limited offline data for representation pre-training. In contrast, LiRE performs substantially worse on Adroit due to its reliance on repeated trajectory segments. These show that FB-PbRL effectively utilizes static human preferences in high-dimensional manipulation tasks. We provide further analysis of human preference noise and synthetic noise ablations in Appendix F.9.

### 4.3. Training efficiency and computational overhead

Beyond asymptotic performance, we investigate the practical training efficiency of our method. Although FB-PbRL involves an initial pre-training phase, it demonstrates exceptionally fast convergence during task-specific fine-tuning. As illustrated in Figure 6, when evaluated on the Walker domain, FB-PbRL reaches competitive performance within roughly 200k steps. Notably, it surpasses the strongest baseline in approximately one hour of fine-tuning wall-clock time, whereas baseline tend to plateau earlier. Therefore, despite a somewhat higher per-step computational cost, FB-PbRL achieves superior performance under equal time budgets and provides substantially faster convergence. A detailed breakdown of the total computational overhead, including the amortized pre-training cost, is provided in Appendix F.6.

### 4.4. Ablation Study

**Efficacy of contrastive learning.** We evaluate whether incorporating a reward model during fine-tuning is sufficient to align representations by comparing against FB-BT-FT, which integrates a pre-trained BT reward model into FB training. Further implementation details are in Ap-

*Table 3.* **Performance on human preference datasets for manipulation.** We evaluate on the Adroit Pen tasks and the MetaWorld Button-Press-Topdown task, all with human-labeled preferences.

| Task | DPPO | LiRE | Ours-FT |
|---|---|---|---|
| Pen-human | **76.3±14.4** | 4.5±1.8 | 70.2±9.8 |
| Pen-cloned | 75.1±7.7 | 14.4±8.2 | **89.0±10.1** |
| Button-Press Topdown | 50.4±9.6 | 70.0±8.8 | **71.2±9.5** |

*Table 4.* **Comparison with FB-BT-FT models.** We report the aggregated performance on *Cheetah*, *Walker*, and *Quadruped*.

| Domain | FB-BT-FT | Ours-FT |
|---|---|---|
| Cheetah | 536.6±42.4 | **621.7±16.1** |
| Walker | 600.6±34.9 | **794.5±31.2** |
| Quadruped | 714.1±17.5 | **846.9±10.7** |

pendix E.2. As shown in Table 4, FB-BT-FT consistently underperforms FB-PbRL (Ours-FT), which directly applies contrastive learning to align the latent geometry with preference structure. This indicates that contrastive learning is beneficial for effective representation alignment, yielding superior generalization and control performance.

**Coefficient $\alpha$ of preference loss.** Figure 5c evaluates sensitivity to the preference coefficient $\alpha$, where setting it to zero corresponds to no contrastive learning. FB-PbRL consistently outperforms this baseline across all coefficients, demonstrating the benefit of contrastive learning. Furthermore, as detailed in Appendix F.7, our method exhibits stable performance across a wide range of values on all Walker tasks, indicating it is not highly sensitive to this hyperparameter A coefficient of 100 yields the strongest and most stable performance and is used as the default in all experiments.

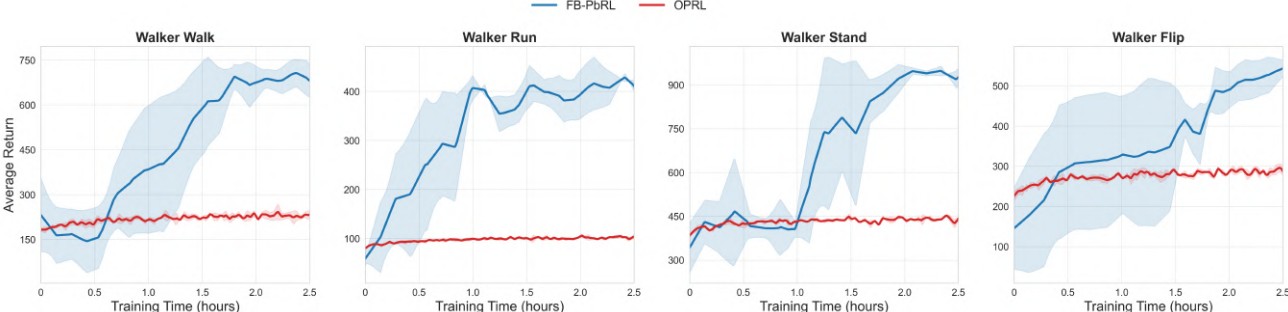

*Figure 6.* **Training efficiency over wall-clock time.** Training curves on the four *Walker* tasks evaluated under equal wall-clock time budgets. FB-PbRL demonstrates highly efficient task-specific fine-tuning, rapidly converging and surpassing the strongest offline PbRL baseline in approximately one hour of training time. Shaded regions denote the standard deviation across 3 seeds.

## 5. Related Work

**Offline PbRL with reward models.** A major line of work combines supervised reward models with offline RL (Christiano et al., 2017; Ouyang et al., 2022), with subsequent advances along three axes. (i) *Feedback efficiency*: LiRE (Choi et al., 2024) improves efficiency via ranked trajectory lists from ternary preferences, while CLARIFY (Mu et al., 2025) selects more informative queries by embedding preference signals into trajectories. When active querying is allowed, OPRL (Shin et al., 2023) applies pool-based active learning to offline datasets. (ii) *Reward over-optimization*: STAR (Bai et al., 2025) introduces preference margin regularization, and DTR (Tu et al., 2025) leverages conditional sequence modeling to mitigate trajectory stitching. Distributionally robust formulations further yield provably efficient model-based (Zhan et al., 2024) and adversarial policy optimization methods (Kang & Oh, 2025). (iii) *Data augmentation*: Data augmentation reduces labeling cost and improves scalability (Kang et al., 2025; Park et al., 2022), while SeqRank (Hwang et al., 2023) demonstrates improved feedback efficiency via sequential preference ranking.

**Offline PbRL without reward models.** Another line of work directly optimizes policies from preferences without explicit reward modeling. OPPO (Kang et al., 2023) learns latent-conditional policies via hindsight matching and contrastive preference modeling, while DPPO (An et al., 2023) adopts a contrastive scoring metric favoring preferred segments. CPL (Hejna et al., 2024) combines contrastive learning with regret-based modeling, and Preference Transformer (Kim et al., 2023) models non-Markovian preferences with transformers. IPL (Hejna & Sadigh, 2023) directly optimizes implicit rewards induced by Q-functions, while FTB (Zhang et al., 2023) uses diffusion models guided by pairwise preferences to generate preferred trajectories.

Due to page limits, we defer discussion of online interactive PbRL and RFRL to Section C.

## 6. Conclusion

In this paper, we reinterpret offline PbRL through the lens of FB-based RFRL. We show that the standard preference loss in PbRL is closely connected to SimCLR-style contrastive learning under FB, motivating the FB-PbRL framework that integrates RFRL pre-training with preference-guided search and fine-tuning. Our results demonstrate that RFRL pre-trained representations provide compact and informative features for PbRL, and we hope this connection encourages further exploration of improving PbRL via RFRL.

While FB-PbRL demonstrates strong empirical performance, its effectiveness still depends on the coverage of the reward-free offline dataset and the reliability of preference supervision. Sparse dataset coverage or highly noisy preferences can weaken preference-guided adaptation, especially in underrepresented regions. One promising direction is to extend the framework to an online setting. The preliminary results in Appendix F.10 suggest that FB-PbRL can be extended to online PbRL by combining online FB representation learning (Sun et al., 2025) with preference-based task specification. In addition, although our main implementation focuses on FB representations, the underlying connection between PbRL and SimCLR-style contrastive learning is not restricted to FB; it can be extended to other RFRL methods with latent linear reward parameterizations, as discussed in Appendix B.3.

## Acknowledgment

This research is partially supported by the National Science and Technology Council (NSTC) of Taiwan under Grant Numbers 114-2628-E-A49-002 and 114-2634-F-A49-002-MBK. This work is also partially supported by the Center for Intelligent Team Robotics and Human-Robot Collaboration under the "Top Research Centers in Taiwan Key Fields Program" of the Ministry of Education (MOE), Taiwan. We also thank the National Center for High-performance Computing (NCHC) for providing computational and storage resources.

## Impact Statement

Preference-based reinforcement learning (PbRL) offers a promising alternative to hand-designed reward functions by learning directly from human feedback, but current offline approaches require substantial labeled preferences. This work introduces an efficient framework that connects reward-free representation learning with PbRL, significantly reducing reliance on costly human annotations. By improving preference efficiency, the proposed method can lower the barrier to deploying RL systems in domains where expert feedback is scarce or expensive, such as robotics, healthcare, and scientific experimentation.

At the same time, reducing the amount of human feedback does not eliminate the need for careful oversight. Learned representations may encode biases present in offline data, and misaligned preferences could still lead to unintended behaviors if deployed without safeguards. We hope this work encourages further research on combining representation learning with preference-based feedback while maintaining transparency, robustness, and responsible human-in-the-loop practices.

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

# Appendix

## A. Pseudo Code of FB-PbRL

Algorithm 2 presents the complete pseudocode of the proposed FB-PbRL framework, integrating the standard Forward-Backward (FB) representation learning with SimCLR contrastive preference learning.

- Lines 4-5 correspond to the standard FB training procedure, where transition batches are sampled from the offline dataset and used to optimize the measure loss and orthonormality regularization.

- Lines 7-19 describe the anchor-based contrastive learning module for preference supervision. Preference pairs are sampled from the preference dataset and encoded into the FB latent space using the backward representation. Ordered latent pairs are constructed based on preference labels, while ties are handled symmetrically by adding both orderings. The resulting set of ordered pairs is then used to compute a SimCLR preference loss, where $\text{sim}(\cdot, \cdot)$ denotes the cosine similarity between two vectors. This objective encourages the learnable anchor latent vector to align with preferred trajectories and repel non-preferred ones.

- Lines 21-25 perform parameter updates. The FB networks are updated using the combined FB objective, while the backward network and the anchor latent vector are additionally optimized with respect to the contrastive preference loss. The policy is updated using the standard TD3-actor objective.

- Finally, the anchor latent vector is normalized to lie on a hypersphere to stabilize optimization.

Below we describe all the loss functions used in the FB-PbRL framework.

**Measure Loss.** The measure loss $\mathcal{L}_{\mathrm{m}}(\theta, \bar{\omega}; \boldsymbol{z}^*)$ minimizes the Bellman residual between the forward representation $\mathbf{F}_\theta$ and the backward representation $\mathbf{B}_\omega$, effectively enforcing Bellman consistency for the implicit successor measures. By projecting these measures onto a learned basis, the objective disentangles the environment's transition dynamics from any specific reward function. This structural separation is crucial as it provides a task-agnostic foundation, enabling the agent to generalize zero-shot to new tasks by understanding the environment's causal structure independent of immediate goals.

$$
\begin{aligned}
\mathcal{L}_{\mathrm{m}}(\theta, \omega; \boldsymbol{z}) := \; & \mathbb{E}_{\substack{(s,a,s') \sim \mathcal{D} \\ (s^\dagger, a^\dagger) \sim \mathcal{D}}} \Big[ \big( \mathbf{F}_\theta(s, a, \boldsymbol{z})^\top \mathbf{B}_\omega(s^\dagger, a^\dagger) - \gamma \, \mathbf{F}_{\hat{\theta}}(s', \pi(s'), \boldsymbol{z})^\top \mathbf{B}_{\hat{\omega}}(s^\dagger, a^\dagger) \big)^2 \Big] \\
& - 2 \, \mathbb{E}_{(s,a,s') \sim \mathcal{D}} \Big[ \mathbf{F}_\theta(s, a, \boldsymbol{z})^\top \mathbf{B}_\omega(s', a') \Big].
\end{aligned}
\tag{19}
$$

**Orthonormality Loss.** To ensure the learned backward representation $\mathbf{B}_\omega$ forms a stable and non-degenerate basis, we impose an orthonormality regularization loss. This constraint acts as a crucial regularizer that prevents representational collapse by encouraging the basis functions to be distinct and mutually orthogonal in expectation. By enforcing the covariance of the backward embeddings to approximate the identity matrix, we ensure that the successor measures are projected onto a diverse and well-conditioned feature space.

$$
\begin{aligned}
\mathcal{L}_{\mathrm{ortho}}(\omega) := \; & \Big\| \mathbb{E}_{(s,a) \sim \mathcal{D}} \big[ \mathbf{B}_\omega(s, a) \mathbf{B}_\omega(s, a)^\top \big] - \mathbf{I}_d \Big\|_{\mathrm{F}}^2 \\
= \; & \mathbb{E}_{\substack{(s,a) \sim \mathcal{D} \\ (s^\dagger, a^\dagger) \sim \mathcal{D}}} \Big[ (\mathbf{B}_\omega(s, a)^\top \mathbf{B}_\omega(s^\dagger, a^\dagger))^2 - \| \mathbf{B}_\omega(s, a) \|_2^2 - \| \mathbf{B}_\omega(s^\dagger, a^\dagger) \|_2^2 \Big].
\end{aligned}
\tag{20}
$$

**Preference Loss.** The contrastive preference loss $\mathcal{L}_{\mathrm{pref}}$ serves a dual optimization role. First, with respect to the task anchor $\boldsymbol{z}$, the loss updates the vector to align with preferred trajectory embeddings, effectively searching for the latent direction that best represents the task. Second, regarding the backward model $\mathbf{B}_\omega$, the loss shapes the latent space geometry, encouraging the representation to map preferred behaviors closer to the task anchor. Standard FB pre-training assumes latent vectors $\boldsymbol{z}$ lie on a unit hypersphere, consistent with the normalization used in SimCLR-style contrastive learning. Let $\text{sim}(\cdot, \cdot)$ denote cosine similarity. We define the preference loss as

$$
\mathcal{L}_{\mathrm{pref}}(\boldsymbol{z}, \omega) := -\mathbb{E}_{\mathcal{D}_{\mathrm{pref}}} \left[ \log \frac{e^{\text{sim}(\boldsymbol{z}, \boldsymbol{z}^+ \sigma(\omega))}}{e^{\text{sim}(\boldsymbol{z}, \boldsymbol{z}^+ \sigma(\omega))} + e^{\text{sim}(\boldsymbol{z}, \boldsymbol{z}^- \sigma(\omega))}} \right],
\tag{21}
$$

where the expectation is over sampled preference pairs $(\sigma^+, \sigma^-)$ from $\mathcal{D}_{\mathrm{pref}}$, and the dependence on $\mathbf{B}_\omega$ is through $\boldsymbol{z}_\sigma^+(\omega)$ and $\boldsymbol{z}_\sigma^-(\omega)$.

---

**Algorithm 2** FB-PbRL

---

1: **Input:** Datsets: Offline dataset $\mathcal{D}$, preference dataset $\mathcal{D}_{\text{pref}}$;
   Hyperparameters: Transition batch size $I$, Preference batch size $J$, discount $\gamma$, Polyak coefficient $\zeta \in [0, 1)$, orthonormality weight $\lambda \in \mathbb{R}_+$, preference weight $\alpha \in \mathbb{R}_+$, learning rates $\mu_{\text{FB}}, \mu_\pi, \mu_z$;
   Pre-trained networks $\mathbf{F}_{\bar{\theta}}, \mathbf{B}_{\bar{\omega}}, \pi_{\bar{\phi}}$ and targets networks $\mathbf{F}_{\hat{\theta}}, \mathbf{B}_{\hat{\omega}}, \pi_{\hat{\phi}}$; Initial anchor $\boldsymbol{z}^* \in \mathbb{R}^d$.
2: **for** $t = 1, 2, \ldots$ **do**
3:     // Standard FB Framework
4:     Sample transition batch $\mathcal{B}_t = \{(s_i, a_i, s_i')\}_{i=1}^I \sim \mathcal{D}$
5:     Compute measure loss $\mathcal{L}_{\text{m}}(\bar{\theta}, \bar{\omega}; \boldsymbol{z}^*)$ ( Equation (16)) and orthonormality loss $\mathcal{L}_{\text{ortho}}(\bar{\omega})$ ( Equation (20)) using $\mathcal{B}_t$
6:     // Anchor-based Contrastive Learning
7:     Sample a batch of preferences $\{(\sigma_j^{(1)}, \sigma_j^{(2)}, y_j)\}_{j=1}^J \sim \mathcal{D}_{\text{pref}}$
8:     Initialize preference pair set $\mathcal{P} \leftarrow \emptyset$
9:     **for** $j = 1$ **to** $J$ **do**
10:         $\boldsymbol{z}_j^{(i)} \leftarrow \mathbf{B}_{\bar{\omega}}(\sigma_j^{(i)})$    for $i \in \{1, 2\}$
11:         **if** $y_j = +1$ **then**
12:             $\boldsymbol{z}^+ \leftarrow \boldsymbol{z}_j^{(1)}, \quad \boldsymbol{z}^- \leftarrow \boldsymbol{z}_j^{(2)}$
13:             $\mathcal{P} \leftarrow \mathcal{P} \cup \{(\boldsymbol{z}^+, \boldsymbol{z}^-)\}$
14:         **else if** $y_j = -1$ **then**
15:             $\boldsymbol{z}^+ \leftarrow \boldsymbol{z}_j^{(2)}, \quad \boldsymbol{z}^- \leftarrow \boldsymbol{z}_j^{(1)}$
16:             $\mathcal{P} \leftarrow \mathcal{P} \cup \{(\boldsymbol{z}^+, \boldsymbol{z}^-)\}$
17:         **else**
18:             $\mathcal{P} \leftarrow \mathcal{P} \cup \{(\boldsymbol{z}_j^{(1)}, \boldsymbol{z}_j^{(2)}), (\boldsymbol{z}_j^{(2)}, \boldsymbol{z}_j^{(1)})\}$
19:         **end if**
20:     **end for**
21:     $\mathcal{L}_{\text{pref}}(\boldsymbol{z}^*, \bar{\omega}) \leftarrow -\frac{1}{|\mathcal{P}|} \sum_{(\boldsymbol{z}^+, \boldsymbol{z}^-) \in \mathcal{P}} \log \frac{\exp(\text{sim}(\boldsymbol{z}^*, \boldsymbol{z}^+)/\tau)}{\exp(\text{sim}(\boldsymbol{z}^*, \boldsymbol{z}^+)/\tau) + \exp(\text{sim}(\boldsymbol{z}^*, \boldsymbol{z}^-)/\tau)}$
22:     // Update networks
23:     $\mathcal{L}_\pi(\bar{\phi}) = \mathbb{E}_{s \sim \mathcal{B}_t} \left[ -\mathbf{F}_{\bar{\theta}}(s, \pi_{\bar{\phi}}(s); \boldsymbol{z}^*)^\top \boldsymbol{z}^* \right]$
24:     $(\bar{\theta}, \bar{\omega}) \leftarrow (\bar{\theta}, \bar{\omega}) - \mu_{\text{FB}} \nabla_{\bar{\theta}, \bar{\omega}} (\mathcal{L}_{\text{m}} + \lambda \mathcal{L}_{\text{ortho}})$
25:     $\bar{\omega} \leftarrow \bar{\omega} - \mu_{\text{FB}} \nabla_{\bar{\omega}} (\alpha \mathcal{L}_{\text{pref}})$
26:     $\bar{\phi} \leftarrow \bar{\phi} - \mu_\pi \nabla_{\bar{\phi}} \mathcal{L}_\pi$
27:     $\boldsymbol{z}^* \leftarrow \boldsymbol{z}^* - \mu_{\boldsymbol{z}} \nabla_{\boldsymbol{z}^*} \mathcal{L}_{\text{pref}}$
28:     $\boldsymbol{z}^* \leftarrow \sqrt{d_{\boldsymbol{z}}} \frac{\boldsymbol{z}^*}{\|\boldsymbol{z}^*\|_2}$
29:     // Update target networks via Polyak averaging
30:     $\hat{\theta} \leftarrow \zeta \hat{\theta} + (1 - \zeta) \bar{\theta}$
31:     $\hat{\omega} \leftarrow \zeta \hat{\omega} + (1 - \zeta) \bar{\omega}$
32:     $\hat{\phi} \leftarrow \zeta \hat{\phi} + (1 - \zeta) \bar{\phi}$
33: **end for**

---

**Policy Loss.** To address continuous control under FB, we train a task-conditioned policy $\pi_\phi$ within an actor-critic framework to maximize the expected return for the inferred task $\boldsymbol{z}$. Specifically, the Q-function for any given task $\boldsymbol{z}$ is recovered via the inner product $Q(s, a; \boldsymbol{z}) = \mathbf{F}_\theta(s, a, \boldsymbol{z})^\top \boldsymbol{z}$. Accordingly, the policy is optimized by minimizing the negative expected Q-values:

$$\mathcal{L}_\pi(\phi; \boldsymbol{z}) = \mathbb{E}_{s \sim \mathcal{D}} \left[ -\mathbf{F}_\theta(s, \pi_\phi(s), \boldsymbol{z})^\top \boldsymbol{z} \right]. \tag{22}$$

This objective drives the agent to select actions that align with the geometric direction specified by $\boldsymbol{z}$.

## B. Further Theoretical Analysis and Generalizations

### B.1. Connecting RFRL with SimCLR Contrastive Learning

We propose to leverage the FB representations pre-trained via RFRL in preference-based learning. Recall from Section 2 that we consider a general class of preference models

$$P_\psi(\sigma^+ \succ \sigma^-) = \mu\big(R_\psi(\sigma^+) - R_\psi(\sigma^-)\big), \tag{23}$$

where $\mu : \mathbb{R} \to [0,1]$ is a monotonically increasing function and $R_\psi(\sigma)$ denotes the total expected reward associated with a trajectory $\sigma$ under the reward function $r_\psi$.

Then, we can rewrite the preference loss as

$$\mathcal{L}_{\mathrm{pref}}(\psi) = -\mathbb{E}_{(\sigma^+,\sigma^-)\sim\mathcal{D}_{\mathrm{pref}}}\left[\log\mu\left(\frac{R_\psi(\sigma^+) - R_\psi(\sigma^-)}{\tau}\right)\right] \tag{24}$$

By the reward parameterization in Equation (9) and definitions in Equations (12) and (14),

$$\mathcal{L}_{\mathrm{pref}}(\psi) = -\mathbb{E}\left[\log\mu\left(\frac{k}{\tau}(\boldsymbol{z}_{\sigma^+} - \boldsymbol{z}_{\sigma^-})^\top\psi\right)\right] \tag{25}$$

where $k$ emerges since $\boldsymbol{z}_\sigma := \frac{1}{k}\sum_{t=1}^{k}\mathbf{B}_{\bar{\omega}}(s_t, a_t)$ is defined as the average backward embedding over a $k$-step segment. Substituting $\psi = \mathbf{H}_{\mathbf{B}}^{-1}\boldsymbol{z}$ ( Equation (11)),

$$\mathcal{L}_{\mathrm{pref}}(z) = -\mathbb{E}\left[\log\frac{\exp(\frac{k}{\tau}\boldsymbol{z}^\top\mathbf{H}_{\mathbf{B}}^{-1}\boldsymbol{z}_{\sigma^+})}{\exp(\frac{k}{\tau}\boldsymbol{z}^\top\mathbf{H}_{\mathbf{B}}^{-1}\boldsymbol{z}_{\sigma^+}) + \exp(\frac{k}{\tau}\boldsymbol{z}^\top\mathbf{H}_{\mathbf{B}}^{-1}\boldsymbol{z}_{\sigma^-})}\right] \tag{26}$$

Here we set the temperature $\tau = k$. Thus, the segment length $k$ acts as a temperature parameter and is absorbed into the logits, yielding Equation (13).

Moreover, in the standard FB pre-training, we usually enforce two additional conditions: (i) We apply orthonomality for the backward representation and have $\mathbf{H}_{\mathbf{B}} = \mathbf{I}_d$. (ii) The FB framework presumes that the latent vector $\boldsymbol{z}$ is on the unit ball. By using these two properties, we conclude that the preference loss exactly matches the SimCLR contrastive loss as

$$\mathcal{L}_{\mathrm{pref}}(\boldsymbol{z}) = -\mathbb{E}\left[\log\left(\frac{\exp(\boldsymbol{z}^\top\boldsymbol{z}_\sigma^+)}{\exp(\boldsymbol{z}^\top\boldsymbol{z}_\sigma^+) + \exp(\boldsymbol{z}^\top\boldsymbol{z}_\sigma^-)}\right)\right]. \tag{27}$$

### B.2. Theoretical Analysis of FB-PbRL

We can directly extend the FB analysis of Touati & Ollivier (2021) to PbRL, showing that the learned $F, B$ together with test-time search for $\boldsymbol{z}$ (Equation (13)) yield near-optimal control under standard conditions.

Let $F(s, a, z), B(s, a)$ be pre-trained representations such that for any reward $r$, $\pi_{z_r}$ is $\epsilon_{FB}$-optimal:

$$\|V^{\pi_{z_r}} - V_r^*\|_\infty \le \epsilon_{FB}, \tag{28}$$

where $z_r := \mathbb{E}[r(s,a)B(s,a)]$, $\pi_{z_r} = \arg\max_a F(s, a, z_r)^\top z_r$, and $V_r^*$ is the optimal value for $r$.

Let $r_\psi$ be the true test-time reward (with optimal value $V^*$), $\hat{r}$ an estimate, and $\pi_{\hat{z}_r} = \arg\max_a F(s, a, \hat{z}_r)^\top \hat{z}_r$.

Define for each k-step preference pair $(\sigma^+, \sigma^-)$

$$\phi(\sigma^+, \sigma^-) := \sum_{i=1}^{k} B(s_i^+, a_i^+) - B(s_i^-, a_i^-), \tag{29}$$

and

$$\Sigma_{\mathcal{D}} := \frac{1}{|D|} \sum_{(\sigma^+, \sigma^-) \in D} \phi \phi^\top \succeq 0, \tag{30}$$

with $\lambda_{\min}(\Sigma_D)$ its smallest eigenvalue.

**Proposition 1.** *For any $\lambda > 0$, with probability at least $1 - \delta$,*

$$\|V^{\pi_{\hat{z}_r}} - V^*\|_\infty = \mathcal{O}\left( \frac{1}{(\lambda_{\min}(\Sigma_D) + \lambda)} \sqrt{\frac{d \log(1/\delta)}{|D|} + \lambda} + \epsilon_{FB} \right). \tag{31}$$

*Proof sketch.* **Step 1:** By Proposition 10 in Touati & Ollivier (2021), we have:

$$\|V^{\pi_{\hat{z}_r}} - V^*\|_\infty \le \frac{2\|r_\psi - \hat{r}\|_\infty}{1 - \gamma} + \epsilon_{FB}. \tag{32}$$

**Step 2:** Bound $\|r_\psi - \hat{r}\|_\infty$. With $r_\psi = B(s, a)^\top \psi$ as in Equation (9) (without loss of generality, let $\|\psi\|_2 \le 1$), Lemma 5.1 in Zhu et al. (2023) gives that MLE via the preference loss in Equation (2) can learn $\hat{\psi}$ that approximates the true $\psi$. That is, for any $\lambda > 0$, with probability at least $1 - \delta$,

$$\|\psi - \hat{\psi}\|_{\Sigma_D + \lambda I} \le C_0 \sqrt{\frac{d \log(1/\delta)}{|D|} + \lambda}, \tag{33}$$

where $C_0$ is a constant.

Since the SimCLR objective in Equation (13) is a reparameterization of Equation (2), its solution satisfies $z_{\text{SimCLR}} = H_B \hat{\psi}$ (essentially finding $\hat{\psi}$). Thus, by Cauchy-Schwarz inequality and the property $|a^\top b| = |(a^\top U^{-1/2})(U^{1/2} b)|$ for $U \succ 0$, we have:

$$\begin{aligned} \|r_\psi - r_{\hat{\psi}}\|_\infty &= \max_{(s,a)} |B(s,a)^\top (\psi - \hat{\psi})| \\ &\le \max_{(s,a)} \|B(s,a)\|_{(\Sigma_D + \lambda I)^{-1}} \cdot \|\psi - \hat{\psi}\|_{\Sigma_D + \lambda I}. \end{aligned} \tag{34}$$

Together, this bound demonstrates that representation sufficiency reduces to preference data coverage $\lambda_{\min}(\Sigma_D)$ and estimation error, yielding near-optimal downstream control. $\square$

### B.3. Generalization to Other RFRL Methods

The SimCLR connection derived in Section 3 is not specific to FB. Rather, it follows from a more general latent linear reward structure that is common in reward-free representation learning.

We illustrate this point using Hilbert representations, or HILP (Park et al., 2024). HILP pre-trains a representation $\phi(s)$ and defines a latent reward over transitions as

$$r_z(s, s') = \tilde{\phi}(s, s')^\top z, \qquad \tilde{\phi}(s, s') := \phi(s') - \phi(s), \tag{35}$$

where $z$ specifies the downstream task. A latent-conditioned policy $\pi(a \mid s, z)$ is then trained to optimize rewards of this form. Thus, test-time adaptation can be viewed as searching for a latent vector $z$ that best matches the target task.

In the PbRL setting, plugging $r_z(s, s')$ into the preference loss (Equation (2)) and following the same derivation as

Equation (13) yields a SimCLR-type objective:

$$\mathcal{L}_{\text{pref,HILP}}(z) = -\mathbb{E}\left[\log\left(\frac{\exp(z^\top \tilde{\phi}^+)}{\exp(z^\top \tilde{\phi}^+) + \exp(z^\top \tilde{\phi}^-)}\right)\right], \tag{36}$$

where

$$\tilde{\phi}^+ := \frac{1}{k}\sum_{t=1}^{k}\tilde{\phi}(s_t^+, s_{t+1}^+), \qquad \tilde{\phi}^- := \frac{1}{k}\sum_{t=1}^{k}\tilde{\phi}(s_t^-, s_{t+1}^-). \tag{37}$$

Then the preference loss can be written as

$$\mathcal{L}_{\text{pref,HILP}}(z) = -\mathbb{E}\left[\log\frac{\exp(z^\top \tilde{\phi}^+)}{\exp(z^\top \tilde{\phi}^+) + \exp(z^\top \tilde{\phi}^-)}\right]. \tag{38}$$

This shows the connection arises from the latent reward structure, rather than FB-specific properties, and thus extends to other RFRL methods with similar parameterizations.

## C. Additional Related Work

### C.1. Online Interactive PbRL

Another line of works in PbRL focuses on designing online interactive RL algorithms with preference feedback. To begin with, PEBBLE (Lee et al., 2021b) proposed to combine unsupervised pre-training and off-policy learning to improve both the sample efficiency and feedback efficiency of PbRL. Subsequently, the PEBBLE framework was extensively benchmarked in various robotic tasks by (Lee et al., 2021a). Moreover, online interactive PbRL has been studied in the following aspects: (i) *Feedback efficiency*: To improve sample efficiency via exploration, RUNE (Liang et al., 2022) design an intrinsic reward by measuring the novelty via disagreement across ensemble of learned reward models. By identifying the potential misalignment between the seemingly informative queries and policy learning, QPA (Hu et al., 2024) proposed the idea of policy-aligned queries to improve both sample and feedback efficiency. Zhu et al. (2025) introduced Proximal Policy Exploration (PPE) to encourage exploration in undersampled policy-proximal regions and strike a balance between in-distribution and out-of-distribution queries. (2) *Noisy preferences*: Given noisy preference feedback, RIME (Cheng et al., 2024) proposed a selection-based discriminator method to dynamically filter out noise and ensure robust training. (3) *Equal preferences*: (Liu et al., 2025) introduced Multi-Type Preference Learning (MTPL), which optimizes the model by promoting similar predicted rewards if the behaviors of two trajectories are annotated as equal preferences. Additionally, online PbRL has been extended beyond the standard single-stage setting. In multi-stage control tasks, for instance, STAIR (Luan et al., 2025) identified the stage alignment issue and proposed a two-step approach that first learns a stage approximation and then prioritizes pairwise comparisons within the same stage. More recently, Guo et al. (2025) utilized human preference feedback as a way to measure the human-likeness of RL agents.

### C.2. Reward-Free Representation Learning

The work most closely related to ours in RFRL includes (Barreto et al., 2017; Borsa et al., 2019; Touati & Ollivier, 2021; Touati et al., 2023). A common reward-free formulation assumes rewards are linear combinations of known features and aims to learn policies that generalize across reward functions. Within this framework, Barreto et al. (2017) introduced successor features (SFs), which encode occupancy measures and generalize the successor representation (Dayan, 1993), enabling efficient policy evaluation and generalized policy improvement. Borsa et al. (2019) further improved SFs by decoupling policy learning from task specification, and later work extended SFs to distributional settings (Zhu et al., 2024).

A key limitation of SF-based methods is their reliance on hand-crafted features. To overcome this, Touati & Ollivier (2021) proposed FB representations, which learn latent features from reward-free offline data via low-rank factorization of optimal policy occupancies. FB representations have been validated in both discrete and continuous control (Touati & Ollivier, 2021; Touati et al., 2023) and extended to a wide range of settings, including offline RL with low-quality data (Jeen et al., 2024; Zheng et al., 2025), online unsupervised RL (Sun et al., 2025), constrained and partially observable RL (Hugessen et al., 2025; Jeen et al., 2025), RL with autoregressive features or unseen dynamics (Cetin et al., 2025; Bobrin et al., 2026), multi-objective RL (Chen et al., 2026), and imitation learning (Pirotta et al., 2024; Tirinzoni et al., 2025). Building on these advances, we reinterpret preference-based RL (PbRL) from a reward-free perspective and connect FB representations to

SimCLR-style contrastive learning to improve feedback efficiency.

In addition to SF- and FB-based methods, RFRL has been studied from various other perspectives. Frans et al. (2024) proposed to learn functional reward encoding (RFE), which encodes the state-reward samples of any arbitrary tasks into latent representations using a transformer-based variational auto-encoder. Agarwal et al. (2025) introduced Proto Successor Measure, which serves as a basis set of representations such that any visitation distribution can be represented as an affine combination of this basis. Wu et al. (2019) learned reward-free representations by considering the MDP graph Laplacian induced by some behavior policy. HILPs (Park et al., 2024) proposed to learn a distance-preserving mapping that maps temporally similar states to spatially similar latent states such that a latent-conditioned policy can be trained to span this latent space by directional movements and then used to solve any downstream tasks. Dubail et al. (2025) showed that low-rank structure could naturally occur in the shifted successor measure, which was defined to capture the environment dynamics after bypassing a number of initial transitions. In contrast, (Wiltzer et al., 2024) focused on the distributional successor measure, which can be learned by minimizing a two-level maximum mean discrepancy. Moreover, to predict the occupancy measure from reward-free data, several recent works leveraged flow matching to learn predictive probabilistic models (Farebrother et al., 2025; Zheng et al., 2026). TD-JEPA (Bagatella et al., 2026) learned latent-predictive representations from reward-free data via temporal difference methods and trained encoders that could capture a low-rank factorization of long-term transition dynamics.

## D. Task and Dataset

### D.1. Task

#### D.1.1. DMCONTROL ENVIRONMENTS

We evaluate our method on 4 standard continuous-control domains from the DeepMind Control Suite (Tassa et al., 2018), including both locomotion and goal-reaching tasks.

**Cheetah.** Cheetah is a planar bipedal locomotion environment with a 17-dimensional continuous state space and a 6-dimensional continuous action space, where each action lies in $[-1, 1]$. The reward is linearly proportional to the forward velocity $v$. We consider four task variants: *Run* and *Walk*, which encourage fast and moderate forward locomotion respectively, as well as *Run Backward* and *Walk Backward*, which reverse the direction of the velocity objective.

**Walker.** A planar walker with a 24-dimensional state space and a 6-dimensional continuous action space bounded in $[-1, 1]$. The environment requires coordinated leg motion and torso balance. We consider multiple task variants, including *Stand*, which encourages maintaining an upright torso, *Walk* and *Run*, which reward increasing levels of forward velocity, and *Flip*, which encourages angular momentum.

**Quadruped.** Quadruped models a four-legged robot has a 78-dimensional continuous state space and a 12-dimensional continuous action space, with each action bounded in $[-1, 1]$. We evaluate four task variants: *Stand*, which encourages stable posture, *Walk* and *Run*, which promote forward locomotion at different speeds, and *Jump*, which requires coordinated leg thrusts to achieve vertical motion.

**Pointmass.** PointMass is a two-dimensional continuous navigation environment consisting of a four-room maze. The state space is 4-dimensional, comprising the position and velocity of the point mass, and the action space is 2-dimensional, corresponding to bounded continuous controls applied in the plane. The initial state is sampled from the top-left room. We evaluate four goal-reaching tasks, where the objective is to move the point mass to a goal located at the center of each room.

#### D.1.2. MANIPULATION ENVIRONMENT.

In addition to locomotion and navigation tasks, we evaluate our method on robotic manipulation environments from MetaWorld and Adroit.

**Button-Press-Topdown.** We focus on *Button-Press-Topdown*, in which a robotic arm is required to press a button from a top-down configuration. During data collection, the button position is randomized across episodes, introducing variation in target location while preserving the underlying task structure.

**Pen.** We further evaluate on the Adroit *Pen* manipulation task, which involves controlling a high-dimensional anthropomorphic hand to reorient a pen to a target pose. We consider two offline datasets: *Pen-Human*, collected from human demonstrations, and *Pen-Cloned*, generated by a behavior-cloned policy. These tasks are characterized by sparse rewards, making them particularly challenging for offline preference-based reinforcement learning methods.

## D.2. Dataset

We use offline datasets from ExORL (Yarats et al., 2022), which consists of datasets collected by several unsupervised reinforcement learning algorithms on the DeepMind Control Suite. In this work, we select datasets generated by Random Network Distillation (RND) (Burda et al., 2019), following prior work on reward-free and zero-shot reinforcement learning (Touati et al., 2023; Park et al., 2024; Agarwal et al., 2025; Jajoo et al., 2025), we use the first 5,000 trajectories from each dataset, where each trajectory contains 1,000 transitions, resulting in a fixed offline dataset of 5 million transitions per environment.

**Preference dataset from scripted teacher.** We construct preference datasets using a scripted teacher based on the ground-truth environment rewards. In the first setting, we randomly sample two trajectories from the offline dataset and extract a continuous segment of length $H = 200$ from each trajectory. Preferences are assigned by comparing the cumulative rewards of the two segments. If the absolute return difference satisfies $|\Delta R| < \epsilon \cdot H \cdot r_{\mathrm{avg}}$. the pair is labeled as a tie. Here, $r_{\mathrm{avg}}$ denotes the average reward of the offline dataset, and we set the threshold coefficient $\epsilon = 0.05$. Using this procedure, we construct a total of 2,000 preference pairs. For PointMass tasks, we do not apply the tie criterion. Since PointMass is a goal-reaching task and the low-quality offline dataset, the reward signal is extremely sparse. As a result, even with a very small $\epsilon$, the above criterion would produce an overwhelming fraction of tie labels, making preference supervision uninformative. We therefore assign deterministic preferences for PointMass by directly comparing the cumulative returns of the two segments.

To ensure a fair comparison with zero-shot reinforcement learning methods that operate on 10,000 transitions, we introduce a second preference construction setting. Specifically, we first sample 400 trajectory segments of length $H = 25$ from the offline dataset, resulting in exactly 10,000 transitions. Preference labels are then generated by pairing these segments using the same scripted teacher and tie threshold, yielding 2,000 preference pairs in total.

**Noisy preference setting.** To evaluate robustness to imperfect preference annotations, we additionally consider a noisy preference setting by injecting synthetic noise into the preference labels (Lee et al., 2021a). Specifically, for each preference pair, we independently corrupt the original label with probability $p$. If the original label indicates a strict preference, it is randomly replaced by either the inverted preference or a tie. Conversely, if the original label indicates a tie, it is converted into a strict preference for either segment with equal probability.

**Preference dataset from human feedback.** For manipulation tasks, we additionally evaluate on human preference datasets without modification, following the original dataset of prior work. For MetaWorld, we use the human preference dataset released with LiRE (Choi et al., 2024), consisting of 200 preference pairs with a segment length of 25. For Adroit *Pen*, we use the human preference dataset from Preference Transformer (Kim et al., 2023), which contains 100 preference pairs with a segment length of 100. In all cases, we directly use the original datasets and codebases without any relabeling or additional preprocessing.

# E. Experimental Details

## E.1. Computation Resource

For all our experiments, we run on RTX 6000 Ada generation, RTX 3080, RTX 3090, and RTX 4080 super GPU. For FB pre-training, it took around 20 hours to train. For FB-PbRL fine-tuning, it took around 10 hours.

## E.2. Implementation Details

**DPPO.** We implement this baseline algorithm based on the open-source code of DPPO (An et al., 2023). The preference predictor provided in the official DPPO repository, which follows a GPT-2 based transformer architecture inherited from the Preference Transformer framework. The hyperparameters used are listed in Table 5.

**OPPO.** We implement this baseline using the official open-source code of OPPO (Kang et al., 2023), and the corresponding hyperparameters are summarized in Table 6. Regarding the dimensionality of the latent task representation $z$, we additionally experimented with increasing its dimension to match that used in our method. However, we did not observe any consistent performance improvement across tasks. Therefore, we report results using the original OPPO configuration.

**CLARIFY & OPRL.** We implement these two baseline algorithm based on the open-source code of CLARIFY (Mu et al., 2025). CLARIFY and OPRL are two-step PbRL methods. In these methods, the reward model is first trained, followed by offline RL using the trained reward model. The reward models used in CLARIFY and OPRL share the same structure. We

*Table 5.* **Hyperparameters of DPPO**

|  | Hyperparameter | Value |
|---|---|---|
| **Preference predictor** | Optimizer | AdamW |
|  | Learning rate | 1e-4 |
|  | Hidden dim | 256 |
|  | Hidden layers | 1 |
|  | $\nu$ | 1.0 |
|  | $m$ | 20 |
| **Policy optimization** | Optimizer | Adam |
|  | Learning rate | 3e-4 |
|  | Hidden dim | 256 |
|  | Hidden layers | 2 |
|  | $\lambda$ | 0.1 |
|  | Dropout | 0.25 |

*Table 6.* **Hyperparameters of OPPO**

|  | Hyperparameter | Value |
|---|---|---|
| **Offline HIM** | $\alpha$ | 0.5 |
|  | $\beta$ | 0.1 |
|  | Optimizer | AdamW |
|  | Training steps | 1e5 |
| **$z^*$ searching** | Optimizer | AdamW |
|  | Learning rate | 1e-3 |
|  | Weight decay | 1e-4 |
|  | Latent dim | 16 |
| **Transformer** | Learning rate | 1e-4 |
|  | Batch size | 256 |
|  | Hidden dim | 128 |
|  | Hidden layers | 3 |
|  | Activation function | ReLU |
|  | Dropout | 0.1 |
|  | Grad norm clip | 0.25 |

use IQL as the default offline RL algorithm. The hyperparameters for reward learning and policy learning are provided in Table 7.

**LiRE.** We implement this baseline algorithm based on the open-source code of LiRE (Choi et al., 2024). The hyperparameter configuration largely follows that of CLARIFY and OPRL (Table 7), as these methods share a similar architecture consisting of a reward model and an IQL policy. The main differences are summarized in Table 8. In addition, LiRE employs a linear score function in the reward model to compute preference scores.

**FB-PbRL and FB-BT-FT.** We implement our algorithm based on the open-source code of FB (Tirinzoni et al., 2025) to pre-train FB model. For the Adroit *Pen* task, to address performance stability, we adopt the TD-JEPA (Bagatella et al., 2026) architecture by incorporating FlowBC and auxiliary encoders into FB to enhance long-term representation learning. For FB-PbRL, we pre-train a single FB model for each domain (e.g., *Walker*) using the corresponding offline dataset for 3 million steps to ensure computational efficiency. This pre-trained model is then shared across all downstream tasks within that domain, significantly reducing the computational cost of multi-task evaluation. During this pre-training phase, we follow established protocols (Touati et al., 2023) to sample the latent variable $z$ using a mixed strategy: 50% of samples are drawn uniformly from a hypersphere of radius $\sqrt{d}$, while the remaining 50% are encoded from state samples as $z = \mathbf{B}(s)$ with $s \sim \mathcal{D}$. For fine-tuning stage, we treat the task embedding $z$ as a learnable parameter that represents the target policy. In each iteration, we sample a batch of preference pairs to update $z$ via the contrastive preference objective. This optimized

*Table 7.* **Hyperparameters of CLARIFY & OPRL**

| | Hyperparameter | Value |
|---|---|---|
| **Reward model** | Structure | Transformer |
| | Optimizer | Adam |
| | Learning rate | 3e-4 |
| | Batch size | 64 |
| | Hidden dim | 256 |
| | Hidden layers | 3 |
| | Activation function | ReLU |
| | Final activation | Tanh |
| | # of ensembles | 3 |
| | Reward from the ensemble models | Average |
| **Contrastive learning** | Latent dim | 16 |
| | Optimizer | AdamW |
| | Learning rate | 1e-3 |
| | Weight decay | 1e-4 |
| | $\lambda_{amb}$ | 0.1 |
| | $\lambda_{quad}$ | 1 |
| | $\lambda_{norm}$ | 0.1 |
| **IQL** | Optimizer | Adam |
| | Hidden dim | 256 |
| | Hidden layers | 2 |
| | Activation function | ReLU |
| | Learning rate | 3e-4 |
| | Batch size | 256 |
| | Discount factor | 0.99 |
| | $\beta$ | 3.0 |
| | $\tau$ | 0.7 |

$z$ is then used to compute the standard FB losses and update the policy, effectively reshaping the latent space to satisfy the specific constraints of the downstream task.

For FB-BT-FT, The BT model is implemented on the open-source code of Uni-RLHF (Yuan et al., 2024). For the Adroit *Pen* task, we observed that the vanilla FB model fails to perform well. To address this, we adopted a TD-JEPA–style approach by introducing the concepts of **FlowBC** and additional encoders to enhance long-term representation learning. For fine-tuning FB with the BT model, we first trained a BT reward model using 2,000 pairs of preference data. This learned reward then guides the FB training for the final 2 million steps (following 1 million steps of warm-up). We utilize the formulation:

$$z_r = \mathbb{E}_{(s,a)\sim\mathcal{D}}\big[\mathbf{B}_\omega(s,a)\,R_{BT}(s,a)\big], \tag{39}$$

and use high quality BT reward to get a z which will reflect the preference of preference dataset so then better than random sample, have the affect of guidance. Following the formulation, the high-quality BT reward enables the model to sample $z$ that better reflects the preferences encoded in the dataset, effectively providing a form of guidance beyond random sampling. Furthermore, we replaced the intrinsic reward in the Q-function update with the BT reward to better align the learning signal with the preference-based objective:

$$\mathcal{L}_{\mathrm{Q}}(\theta) = \mathbb{E}_{(s,a,s')\sim\mathcal{D}}\Big[\big(\mathbf{F}_\theta(s,a,\boldsymbol{z})^\top\boldsymbol{z} - (R_{BT}(s,a) + \gamma\,\mathbf{F}_{\hat\theta}(s',\pi_\phi(s'),\boldsymbol{z})^\top\boldsymbol{z})\big)^2\Big] \tag{40}$$

In the original FB implementation, the coefficient of this Q-function loss is typically set to zero, meaning that the Q-loss is not utilized during training. In contrast, our formulation explicitly incorporates the BT reward into this objective, enabling the value function to learn from preference-aligned feedback rather than relying solely on intrinsic signals. For FB-PbRL, we fine-tune the pre-trained FB model using preference dataset. The hyperparameters used are listed in Table 9.

*Table 8.* **Hyperparameters specific to LiRE**

| Hyperparameter | Value |
|---|---|
| Reward model score function | Linear |
| Learning rate | 1e-3 |
| Batch size | 256 |
| RLT feedback limit $Q$ | 100 |

## F. Additional Experiment Results

### F.1. Ablation Study of Segment Size

Segment size is an important design choice in constructing preference datasets. Intuitively, longer segments may provide richer context and could enable the FB representation to encode task-relevant information more accurately in the latent space. Based on this intuition, we initially expected larger segment sizes to yield improved performance.

However, empirical results in Table 10 indicate that FB-based methods are relatively insensitive to the choice of segment size. Across Walker tasks, segment sizes of 200 and 25 achieve comparable performance in most cases. One possible explanation is that locomotion tasks often exhibit repetitive and temporally consistent behaviors, such that even shorter segments are sufficient to capture the essential dynamics and preference-relevant information.

### F.2. Comparison Between SimCLR Loss and Margin Loss

We compare our SimCLR contrastive objective with a margin-based contrastive loss commonly used in PbRL methods such as OPPO (Kang et al., 2023) and CLARIFY (Mu et al., 2025).

For the margin loss, we follow the design adopted in CLARIFY (Mu et al., 2025) and implement a triplet margin loss with $\ell_2$ distance. Specifically, we construct positive and negative samples based on preference supervision and apply a standard triplet margin loss with margin $m = 1.0$. In addition, we include an auxiliary triplet loss with zero margin to handle tied or ambiguous preference cases.

Empirically in Table 11, across all three evaluated domains, the SimCLR-based objective consistently outperforms the margin-based loss. We further observe that the margin-based loss exhibits unstable training behavior in several environments, leading to high variance and degraded performance. These results suggest that the SimCLR loss is better suited for structuring the FB latent space.

### F.3. Limited and Noisy Preference on *Walker*

We further investigate the robustness of our method under limited and noisy preference supervision on the *walker* domain. It is important to note that these two settings follow different data construction protocols. For the limited-preference experiments, we adopt the zero-shot RFRL protocol. In contrast, the noisy-preference experiments follow the PbRL protocol, which results in a different initial performance scale.

Overall, the *walker* domain exhibits robustness to both limited and noisy preference supervision that is consistent with the trends observed in *quadruped*. Under limited preferences, performance degrades gradually as the number of preference pairs decreases. When fewer than 500 preference pairs are available, our method falls slightly below the PSM baseline; however, the performance gap remains within 10% across all evaluated settings, indicating a graceful degradation behavior. In the presence of noisy preferences, performance on the *walker* domain remains largely stable as the noise ratio increases, with a noticeable degradation only emerging at a noise ratio of 0.2. This suggests that the learned representation and preference-guided finetuning procedure can effectively tolerate moderate levels of label noise in this domain as well.

Taken together, these results demonstrate that the robustness properties observed in the main text are not specific to a single domain. Instead, similar behavior is also observed on *walker*, providing additional evidence for the robustness of our work.

### F.4. Linear Realizability Rewards under Backward Model

As stated in Equation 9, we assume the linear realizability of the reward function under the backward representation. This implies that the learned latent space is sufficiently rich to express arbitrary downstream task rewards via linear projections.

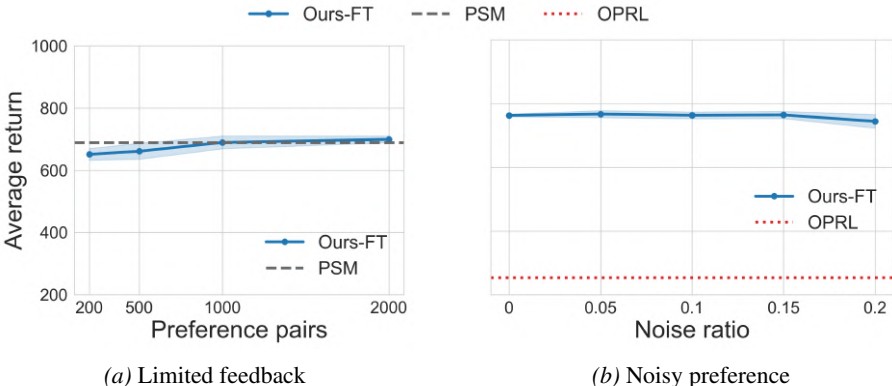

*(a)* Limited feedback                         *(b)* Noisy preference

*Figure 7.* **Robustness analysis of FB-PbRL in the *Walker* domain. (a)** Performance scaling with respect to the number of preference pairs. **(b)** Robustness against noisy preference annotations. Results in (a) and (b) represent the average performance averaged over four tasks in the *Walker* domain; the dashed lines indicate the best performance achieved by offline PbRL and zero-shot RFRL baselines. In all plots, shaded regions indicate the standard deviation across 5 seeds.

To empirically validate this hypothesis, we conduct an experiment to quantify how accurately ground-truth rewards can be reconstructed from frozen backward features. Specifically, We sample 10,000 transitions from the offline RND dataset and solve a linear regression problem to obtain the optimal linear coefficient vector $\mathbf{w}$ that minimizes the Mean Squared Error (MSE):

$$\min_{\mathbf{w}} \mathbb{E}_{(s,a) \sim \mathcal{D}} \left[ \left( r(s,a) - \mathbf{B}_\omega(s,a)^\top \mathbf{w} \right)^2 \right]. \tag{41}$$

.

Table 15 reports the MSE across four domains: *Cheetah*, *Walker*, *Quadruped*, and *Pointmass*. The results demonstrate extremely low reconstruction errors across all tasks. Notably, in the *Pointmass* domain, the MSE is negligible, indicating a near-perfect recovery of the reward landscape. Even in high-dimensional locomotion tasks like *Quadruped* and *Cheetah*, the low MSE values confirm that the unsupervised FB objective has successfully learned a basis that covers the underlying structure of these tasks. These findings strongly support Equation 9, confirming that our pre-trained model yields a versatile task-agnostic representation capable of adapting to downstream tasks via linear operations.

### F.5. Empirical Invertibility of $\mathbf{H_B}$

In our framework, the invertibility of $\mathbf{H_B}$ is inherently maintained by the FB objective, eliminating the need for singularity handling. The auxiliary orthonormality loss regularizes $\mathbf{H_B}$ to approximate the identity matrix ($I_d$). This actively prevents feature collapse and pushes $\mathbf{H_B}$ away from singularity. We empirically validated this stability following the feature rank protocol introduced by (Touati et al., 2023). By evaluating $\mathbf{H_B}$ during fine-tuning across all four walker tasks, we confirm all eigenvalues remained strictly above the $10^{-4}$ threshold. Thus, $\mathbf{H_B}$ successfully maintains full rank and strict invertibility in practice.

### F.6. Computational Overhead

To explicitly quantify the computational overhead, Table 12 details the total computation time required for the 4 tasks in the Walker domain. Our framework separates an amortized representation pre-training phase from task-specific fine-tuning. Because the FB pre-training is performed only once per domain and subsequently reused across all downstream tasks, its effective pre-training cost scales favorably. This provides a distinct advantage over standard end-to-end PbRL methods that require training a new reward model from scratch for every task.

### F.7. Extended Sensitivity Analysis of Hyperparameters

**Sensitivity to preference coefficient $\alpha$.** We additionally evaluate all four tasks within the Walker domain across a wide range of $\alpha$ values. As shown in Table 13, FB-PbRL demonstrates highly stable performance across different magnitudes of the coefficient. This indicates that our method is not overly sensitive to this specific hyperparameter. We select $\alpha = 100$ as the default value across all experiments since it consistently yields the strongest results.

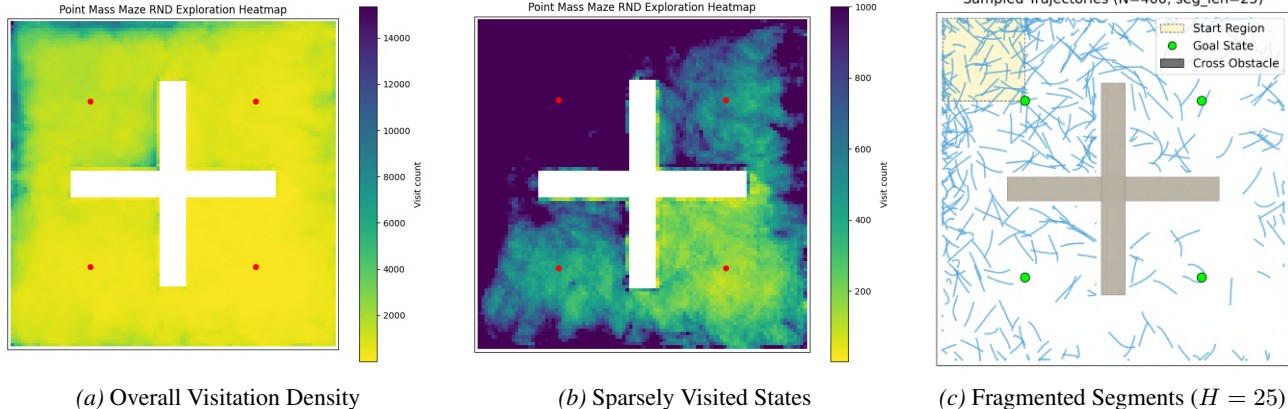

*(a)* Overall Visitation Density     *(b)* Sparsely Visited States     *(c)* Fragmented Segments ($H = 25$)

*Figure 8.* **Dataset coverage and in the PointMass domain.** The layout of the maze is centered around a cross-shaped obstacle (white/empty region), with the dots representing the four target goal states. **(a)** The state visitation heatmap of the RND PointMass dataset reveals a heavy bias toward the top-left region where the agent is initialized. **(b)** Analysis of sparsely visited states shows that the *bottom-right* goal area is severely underrepresented in the offline data, leading to a performance bottleneck. **(c)** Visualization of trajectories with a short horizon of $H = 25$; these fragmented segments provide weaker preference signals, contributing to the higher variance observed in policy performance.

**Sensitivity to orthonormality coefficient $\lambda$.** For the orthonormality coefficient $\lambda$, we set $\lambda = 1$ following the original FB pre-training setup (Touati et al., 2023). To validate this choice within our preference-based framework, we additionally evaluate $\lambda \in \{0.3, 3\}$ on the *Walker-walk* and *Quadruped-walk* tasks. As presented in Table 14, the results show that $\lambda = 1$ consistently achieves the best performance, confirming the prior findings and validating our default setting.

### F.8. Further Analysis on PointMass Tasks

In this section, we provide a detailed analysis of the performance characteristics observed in the *PointMass* domain, specifically addressing the performance gap in the *Bottom Right* task (Table 1) and the variance reported in Table 2.

**Dataset Distribution and Coverage Imbalance.** The performance of FB-PbRL on specific goals is closely tied to the state coverage of the offline dataset. As visualized in the visitation heatmap (Figure 8a), the RND PointMass dataset exhibits a severe **coverage imbalance**, where transitions are heavily concentrated in the top-left region of the maze. Conversely, the bottom-right region contains significantly fewer samples (Figure 8b). This distributional sparsity leads to insufficient supervision during the representation learning phase, making it inherently more challenging to resolve task objectives in these underrepresented areas compared to well-sampled regions.

**Informativeness of Preference Signals.** The zero-shot evaluation protocol poses an additional challenge by restricting the preference budget to only 10k transitions with a short segment length of $H = 25$. Figure 8c visualizes sample segments from this setting. The trajectories are notably short and fragmented, which often fails to capture sufficient state transitions to convey a clear directional signal toward the goal. The combination of data scarcity in the goal region and the weak preference signals from short segments accounts for the increased variance and the difficulty in achieving peak performance in certain navigation tasks.

### F.9. Analysis of Human Preference Noise

We quantify the label noise in the Adroit human preference datasets by measuring the fraction of preferences that contradict ground-truth reward signals. Our analysis reveals that human annotations in these tasks are exceptionally noisy, with noise ratios of 60% for *pen-human* and 55% for *pen-cloned*. To validate the impact of such noise, we conducted a controlled study on the *Quadruped* tasks by injecting synthetic noise into preference labels.

As shown in Figure 9, once the noise ratio approaches 50%, the performance of all preference-based methods drops, and the margin of improvement provided by FB-PbRL shrinks. This confirms that our method's performance in Adroit is primarily bottlenecked by the inherent quality of the human-labeled data.

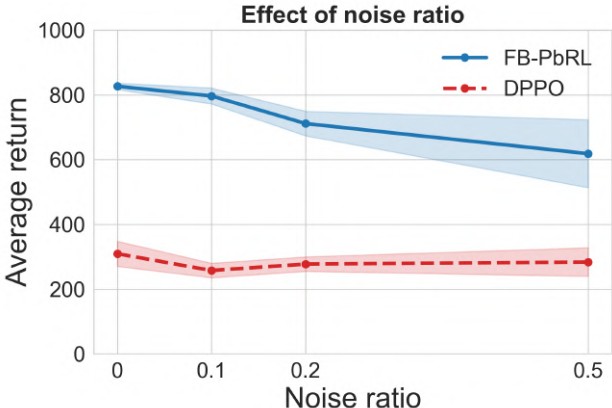

*Figure 9.* **Robustness analysis against preference noise in the *quadruped* domain.** The plot illustrates the performance of FB-PbRL and the DPPO baseline as the preference noise ratio increases. Result is averaged over four *Quadruped* tasks. In all plots, shaded regions indicate the standard deviation across 5 seeds.

### F.10. Extension to Online PbRL Settings

While the primary focus of this work is on the offline regime, our framework can naturally extend to online PbRL settings. This extension is achieved by collecting environment trajectories utilizing an online Forward-Backward representation backbone (Sun et al., 2025), followed by standard preference querying to specify the latent task vector.

To validate this capability, we compare the against a standard online PbRL baseline, PrefPPO (Christiano et al., 2017; Lee et al., 2021a), which learns a reward model from pairwise trajectory preferences and then optimizes the policy with Proximal Policy Optimization with a clipped objective (Schulman et al., 2017; Huang et al., 2024). As presented in Table 16, online FB-PbRL demonstrates consistent and significant performance gains across all evaluated tasks in the *Quadruped* domain. These results confirm that our representation-driven approach extends effectively beyond the offline setting and maintains highly competitive performance in online environments.

### F.11. Training Curve

**Main results.** The following training curves present the main results, showing the performance of FB-PbRL compared with other baselines in the *Cheetah*, *Walker*, and *Quadruped* domains, as well as a comparison between the PbRL and zero-shot RFRL protocols. For the OPRL and CLARIFY baselines, we report training curves up to $2 \times 10^5$ steps, as these methods are originally configured to train for this horizon and exhibit no further performance improvement beyond this point in our experiments. All the corresponding training curves are shown in Figures 10, 11, and 12.

**Robustness** We show the training curve of FB-PbRL under both limited and noisy preference supervision. Under the limited-preference setting, experiments follow the zero-shot RFRL protocol, where dataset coverage is inherently small. As a result, training curves exhibit higher variance, and a gradual performance drop is observed in later stages, which is consistent with the increased sensitivity to data scarcity in this regime.

For noisy preference annotations, we observe a pronounced domain-dependent effect. In particular, performance on the *Quadruped* domain degrades more noticeably as the noise ratio increases, whereas *Walker* remains largely robust and only shows a clear performance drop at a noise ratio of 0.2. All the corresponding training curves are shown in Figure 13, 14.

**Contrastive coefficient.** We observe that setting the contrastive coefficient to 10 leads to degraded performance on *Quadruped-Walk*, suggesting insufficient alignment to capture a meaningful task embedding in this setting. In contrast, larger values as well as the default choice (100) consistently yield strong performance across tasks, indicating that a sufficiently strong contrastive signal is necessary for reliable task embedding learning. The corresponding training curves are shown in Figure 15.

**Segment size.** Varying the segment size has limited impact on most tasks, with performance remaining comparable across different choices. An exception is *Walker-Walk* with segment size 25, which exhibits a noticeable drop in performance. This suggests that Walker tasks benefit from longer temporal context, whereas Quadruped tasks are less sensitive to segment length. The corresponding training curves are shown in Figure 16.

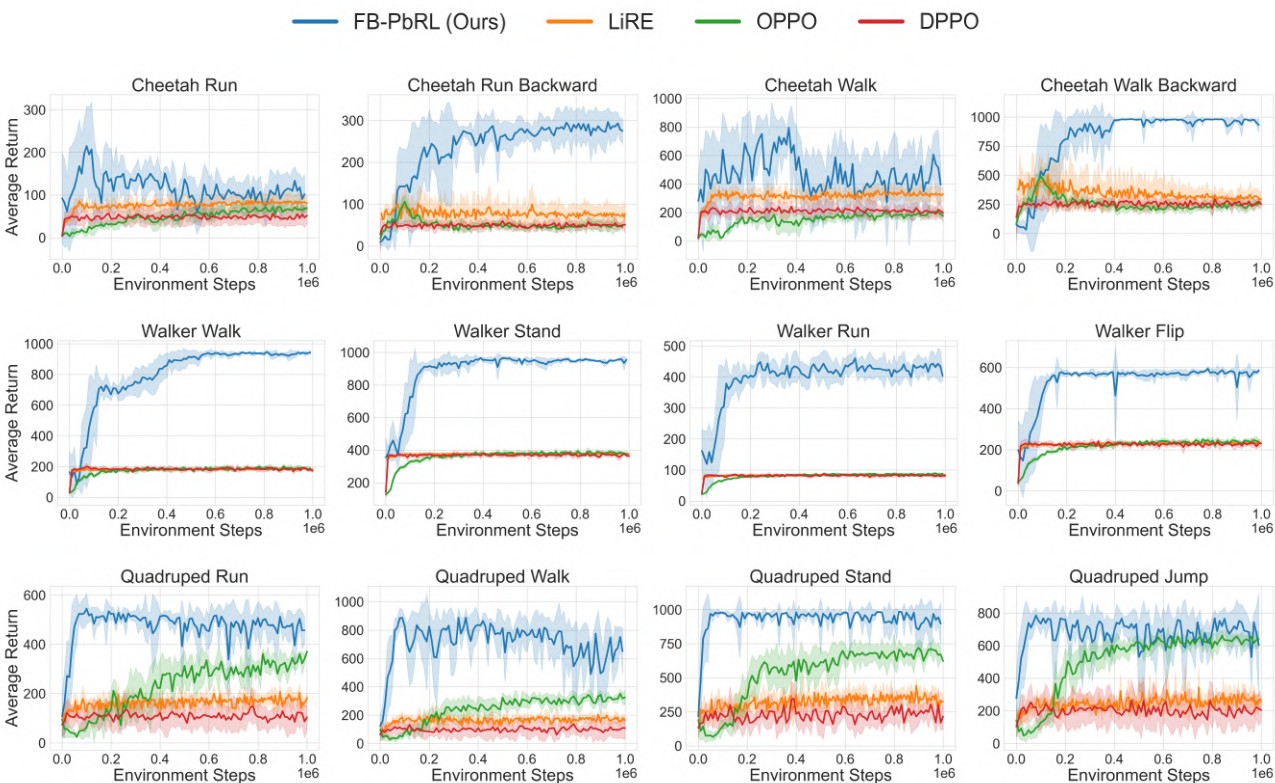

*Figure 10.* **FB-PbRL compared with LiRE, OPPO, and DPPO.** In all plots, shaded regions indicate the standard deviation across 5 seeds.

**Contrastive objective.** We further compare our SimCLR-style contrastive learning objective with a margin-based alternative. Across nearly all tasks, the margin-based objective results in inferior and less stable performance, highlighting the advantage of the SimCLR formulation adopted in our method. The corresponding training curves are shown in Figure 17.

# G. Negative Results

## G.1. Latent-Space Regularization and Consistency Constraints

**Regularization Loss.** A key challenge in offline reinforcement learning is mitigating OOD effects during optimization (Kumar et al., 2020; Kostrikov et al., 2022). While prior FB-based methods typically address this issue by regularizing the actor (Jeen et al., 2024; Zheng et al., 2025), in our setting we observed that OOD behavior can also arise in the latent task space when optimizing the anchor $z$ via preference supervision. To enforce this, we experimented with an auxiliary regularization term, $\mathcal{L}_{\text{reg}} = \|\mathbf{z} - \bar{\mathbf{z}}_{\text{pos}}\|_2^2$, where $\bar{\mathbf{z}}_{\text{pos}}$ denotes the centroid of the positive samples. This term was designed to anchor $\mathbf{z}$ near the centroid of positive preference samples.

However, this explicit regularization yielded no performance gains. We attribute this to the implicit regularization provided by our preference-guided fine-tuning . The contrastive objective inherently reshapes the latent geometry, naturally confining $\mathbf{z}$ to the valid task manifold. Consequently, additional geometric constraints proved redundant and occasionally restrictive compared to the reshaping induced by fine-tuning alone.

**Reconstruction Loss.** The original FB framework does not explicitly enforce a self-consistency constraint in the latent space, i.e., trajectories generated by a policy conditioned on a task vector $z$ may not map back to the same latent representation when re-encoded by the backward model. To address this, we explored a reconstruction-based regularization that encourages consistency between the anchor $z$ and the latent representation $\hat{z}$ obtained by encoding trajectories generated under $z$.

Concretely, we generate multiple trajectories using the actor conditioned on $z$, encode them back into latent representations

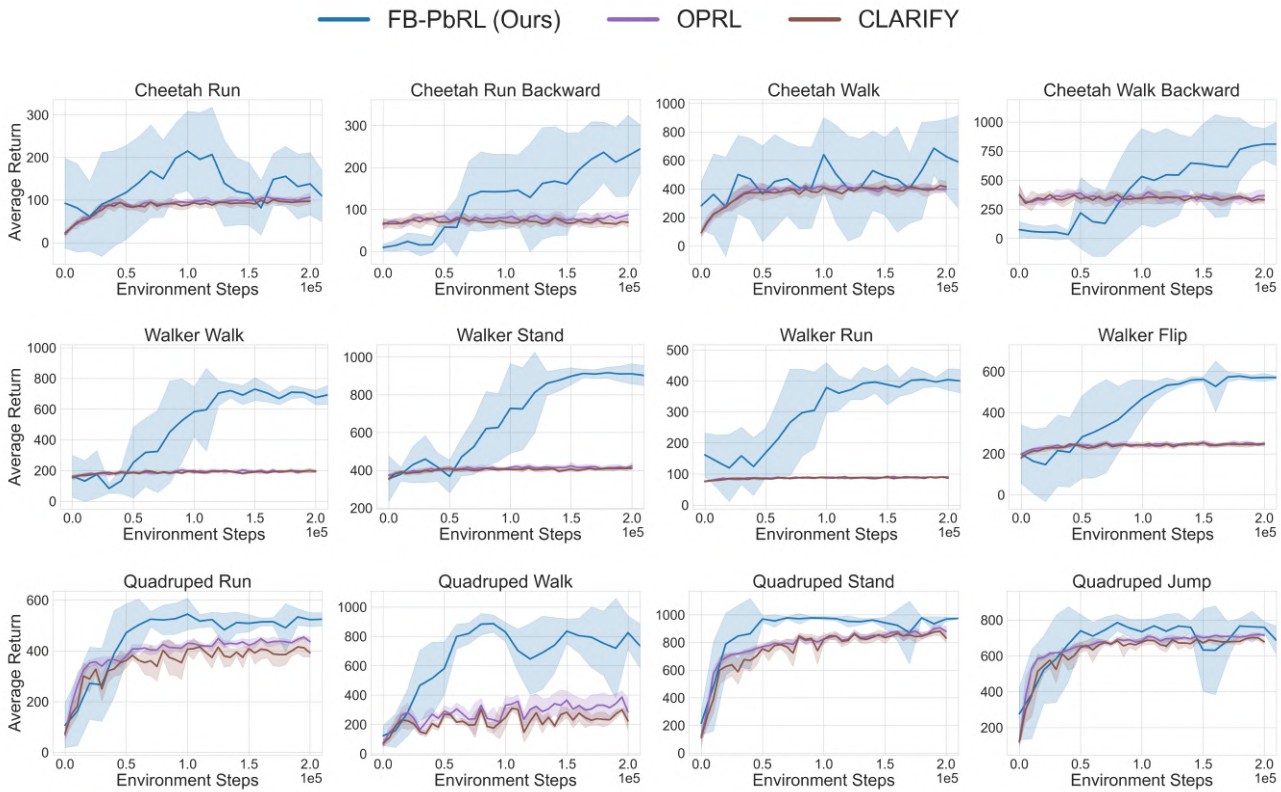

*Figure 11.* **FB-PbRL compared with OPRL and CLARIFY.** In all plots, shaded regions indicate the standard deviation across 5 seeds.

via the backward encoder, and minimize the cosine distance

$$\mathcal{L}_{\text{recon}} = 1 - \cos(\boldsymbol{z}, \hat{\boldsymbol{z}}).$$

However, this approach incurs substantial computational overhead, as it requires repeated environment rollouts during training. In practice, incorporating this loss significantly increases training time and is therefore not feasible in our setting.

### G.2. Mitigating Fast Convergence of the Preference Loss

We observed that the optimization of the task anchor $\boldsymbol{z}$ could converge rapidly under the preference loss. To mitigate this and encourage broader exploration, we experimented with two strategies:

**Multi-Anchor Initialization.** Initializing four anchor vectors and randomly sampling one of them at each update, aiming to diversify the search trajectory.

**Noise Injection.** We added Gaussian noise to $\boldsymbol{z}$ during sampling to induce local exploration around the current estimate.

Contrary to our expectations, neither strategy yielded higher asymptotic performance. While these methods successfully slowed down the convergence rate, they merely delayed the time required to reach peak performance without finding better solutions than the baseline. In contrast, we find that a simple reduction of the learning rate for $\boldsymbol{z}$ is the most effective and stable way to mitigate overly fast convergence of the preference loss, without introducing additional optimization complexity.

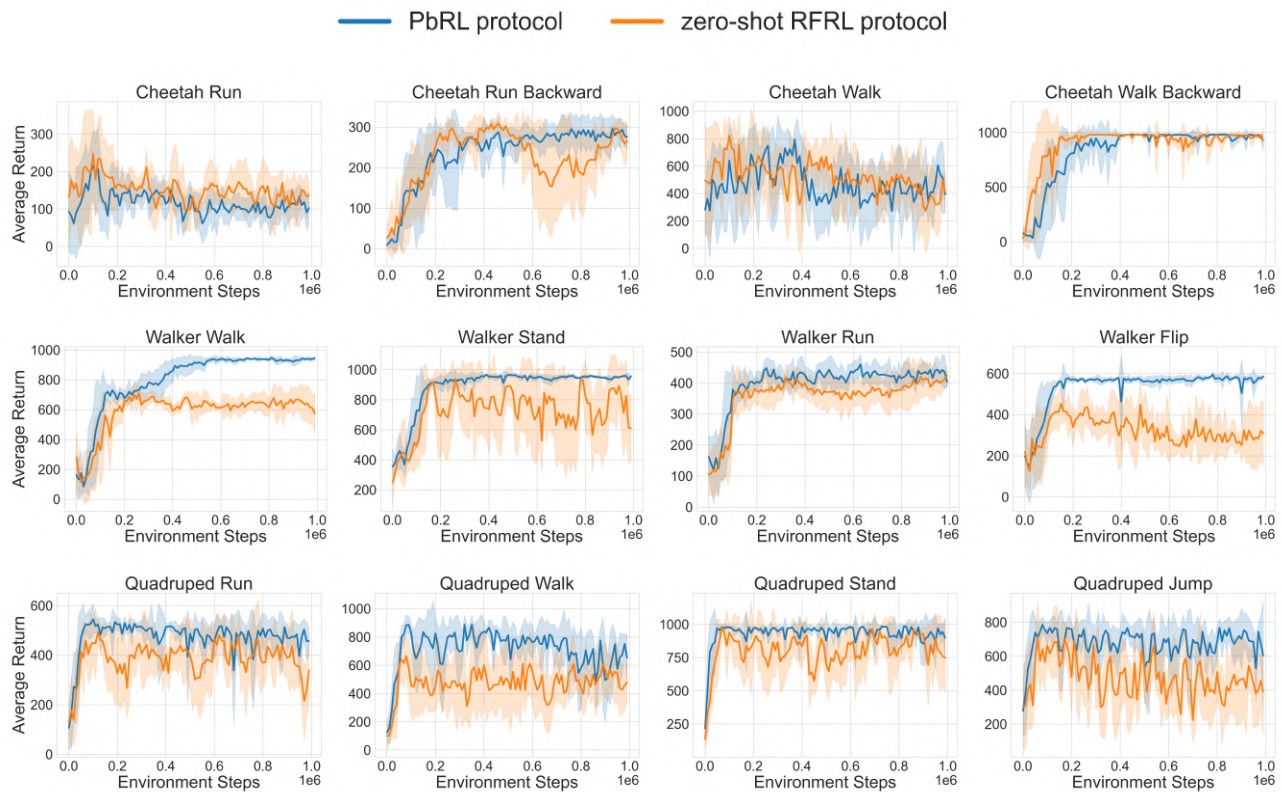

*Figure 12.* **Comparison between the PbRL and zero-shot RFRL protocols.** In all plots, shaded regions indicate the standard deviation across 5 seeds.

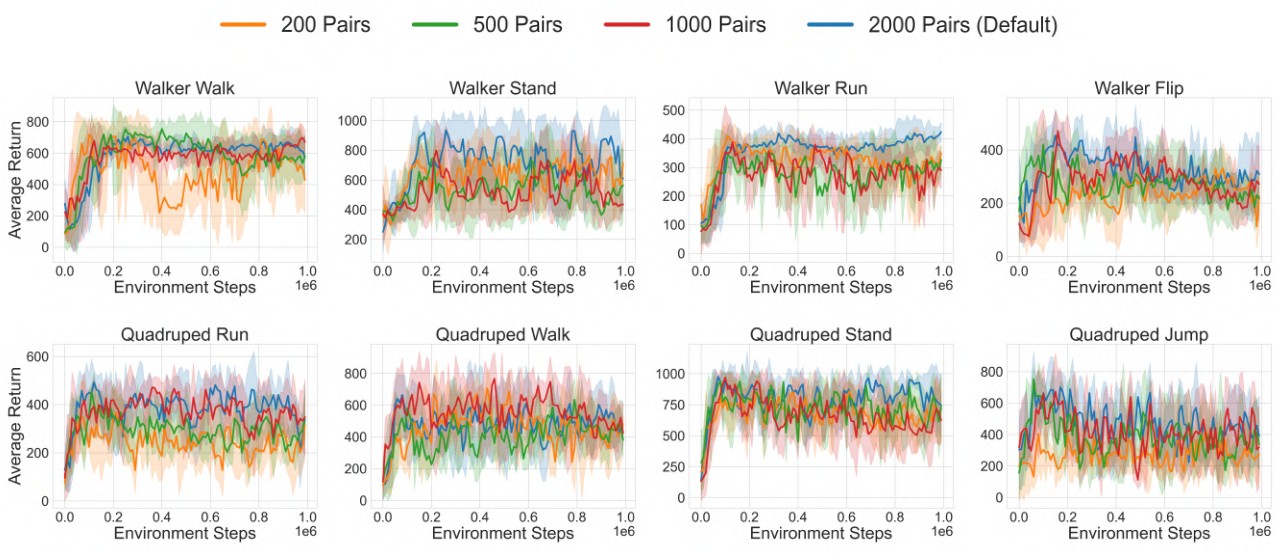

*Figure 13.* **Comparison of FB-PbRL under different numbers of preference pairs.** In all plots, shaded regions indicate the standard deviation across 5 seeds.

*Table 9.* **Hyperparameters of FB reward learning and policy learning**

|  | Hyperparameter | Value |
|---|---|---|
| **Reward model** | Structure | MLP |
|  | Optimizer | Adam |
|  | Learning rate | 3e-4 |
|  | Batch size | 64 |
|  | Hidden dim | 256 |
|  | Hidden layers | 3 |
|  | Activation function | ReLU |
|  | Final activation | Tanh |
|  | # of ensembles | 3 |
|  | Reward from the ensemble models | Average |
| **FB** | Pre-train steps | 3e6 |
|  | Latent dimension | 100 (Walker, PointMass), 50 (others) |
|  | Forward model hidden dim | 1024 |
|  | Backward model hidden dim | 256 |
|  | Actor hidden dim | 1024 |
|  | Forward model hidden layers | 1 |
|  | Backward model hidden layers | 2 |
|  | Actor hidden layers | 1 |
|  | Batch size | 1024 |
|  | Learning rate (Forward) | 1e-4 |
|  | Learning rate (Backward) | 1e-6 (PointMass), 1e-4 (others) |
|  | Learning rate (Actor) | 1e-6 (PointMass), 1e-4 (others) |
|  | Discount factor | 0.99 (PointMass), 0.98 (others) |
|  | $\zeta$ | 0.01 |
|  | $\lambda$ | 1 |
|  | $z$ train-goal ratio | 0.5 |
|  | BC coefficient (for *Pen*) | 0.3 |
|  | Left encoder hidden dim (for *Pen*) | 512 |
| **FB-BT-FT** | Warmup steps | 1e6 |
|  | Fine-tuning steps | 2e6 |
|  | Learning rate | 1e-4 |
|  | Q loss coefficient | 0.1 |
| **FB-PbRL** | Fine-tuning steps | 1e6 |
|  | Learning rate($z$) | 1e-5 |
|  | Sample preference batch size | 256 |
|  | Preference coefficient | 100 |
|  | $\beta$ | 1.0 |

*Table 10.* **Effect of segment size on performance (Walker tasks).** Segment size 200 is used as the default setting. Results for other segment sizes are averaged over 3 random seeds.

| Domain | Task | 200 (default) | 100 | 50 | 25 |
|---|---|---|---|---|---|
| **Walker** | walk | **961.5±4.2** | 950.5±12.1 | 936.5±26.1 | 873.5±18.1 |
|  | stand | 980.3±3.4 | **982.4±1.0** | 987.0±3.1 | 977.5±1.1 |
|  | run | **503.7±14.4** | 497.2±12.2 | 491.4±31.3 | 495.0±24.2 |
|  | flip | 606.0±7.9 | **805.9±34.1** | 664.8±54.3 | **758.1±23.8** |
|  | *Average* | 762.9 | **809.0** | 769.9 | **776.0** |

*Table 11.* **Performance comparison between margin-based loss and SimCLR loss.** Results are averaged over 5 random seeds (Mean $\pm$ Std).

| Domain | Task | Margin Loss | SimCLR Loss |
|---|---|---|---|
| Cheetah | Run | 223.2$\pm$10.1 | **249.5$\pm$74.5** |
| | Run Backward | 333.4$\pm$25.3 | **333.9$\pm$28.5** |
| | Walk | 807.6$\pm$60.3 | **920.3$\pm$30.7** |
| | Walk Backward | 981.6$\pm$1.4 | **983.3$\pm$0.5** |
| | *Average* | 586.5 | **621.7** |
| Walker | Walk | 843.2$\pm$98.2 | **961.5$\pm$4.2** |
| | Stand | 933.2$\pm$72.6 | **980.3$\pm$3.4** |
| | Run | **511.3$\pm$12.6** | 503.7$\pm$14.4 |
| | Flip | 574.4$\pm$85.0 | **606.0$\pm$7.9** |
| | *Average* | 715.5 | **762.9** |
| Quadruped | Run | 525.5$\pm$18.5 | **589.9$\pm$36.3** |
| | Walk | 752.3$\pm$94.6 | **944.2$\pm$8.9** |
| | Stand | 979.9$\pm$6.8 | **991.1$\pm$0.3** |
| | Jump | 757.5$\pm$15.9 | **862.2$\pm$19.8** |
| | *Average* | 753.8 | **846.9** |

*Table 12.* **Total computation time (in hours) for 4 tasks in the Walker domain.** Evaluated using NVIDIA RTX 3090 GPUs. FB-PbRL benefits from amortized pre-training, which is performed only once per domain and reused across all tasks.

| Phase | OPRL | CLARIFY | LiRE | FB-PbRL (Ours) |
|---|---|---|---|---|
| Pre-training / Reward Model Training | 24 | 24 | - | 18.5 |
| Fine-tuning / Offline RL (1M steps) | 20 | 20 | 18 | 50 |

*Table 13.* **Effect of preference coefficient $\alpha$ on performance (Walker tasks).** The method exhibits stable performance across different scales of $\alpha$, with $\alpha = 100$ yielding the strongest results across all tasks. Results are averaged over 3 random seeds.

| Task | $\alpha = 10$ | $\alpha = 100$ (default) | $\alpha = 1000$ |
|---|---|---|---|
| walk | 888.0 | **961.5** | 923.9 |
| stand | 970.9 | **980.3** | 978.2 |
| run | 491.5 | **503.7** | 469.3 |
| flip | 584.1 | **606.0** | 578.7 |

*Table 14.* **Effect of orthonormality coefficient $\lambda$ on performance.** We compare different values of $\lambda$, demonstrating that the default setting of $\lambda = 1$ consistently achieves the best performance. Results are averaged over 3 random seeds.

| Task | $\lambda = 0.3$ | $\lambda = 1$ (default) | $\lambda = 3$ |
|---|---|---|---|
| Walker-walk | 898.6 | **961.5** | 890.6 |
| Quadruped-walk | 932.0 | **944.2** | 925.5 |

*Table 15.* **Linear realizability of ground-truth rewards.** Mean Squared Error (MSE) between the true environment rewards and the rewards predicted via linear regression on the frozen backward representations.

| Domain | Task | MSE |
|---|---|---|
| Cheetah | Run | 0.0029 |
| | Run Backward | 0.0031 |
| | Walk | 0.0576 |
| | Walk Backward | 0.0555 |
| Walker | Walk | 0.0186 |
| | Stand | 0.0236 |
| | Run | 0.0018 |
| | Flip | 0.0277 |
| Quadruped | Run | 0.0061 |
| | Walk | 0.0516 |
| | Stand | 0.0135 |
| | Jump | 0.0162 |
| Pointmass | Top Left | 0.0036 |
| | Top Right | 0.0015 |
| | Bottom Left | 0.0010 |
| | Bottom Right | 0.0004 |

*Table 16.* **Performance comparison in the online PbRL setting.** Results are evaluated on the *Quadruped* domain. Online FB-PbRL significantly outperforms the standard PrefPPO baseline across all tasks.

| Task | PrefPPO | Online FB-PbRL (Ours) |
|---|---|---|
| walk | $279.7 \pm 8.7$ | $\mathbf{597.2 \pm 60.0}$ |
| run | $198.6 \pm 26.3$ | $\mathbf{425.9 \pm 77.8}$ |
| stand | $357.4 \pm 4.6$ | $\mathbf{958.5 \pm 1.2}$ |
| jump | $298.9 \pm 21.3$ | $\mathbf{679.2 \pm 41.5}$ |
| *Average* | 283.7 | **665.2** |

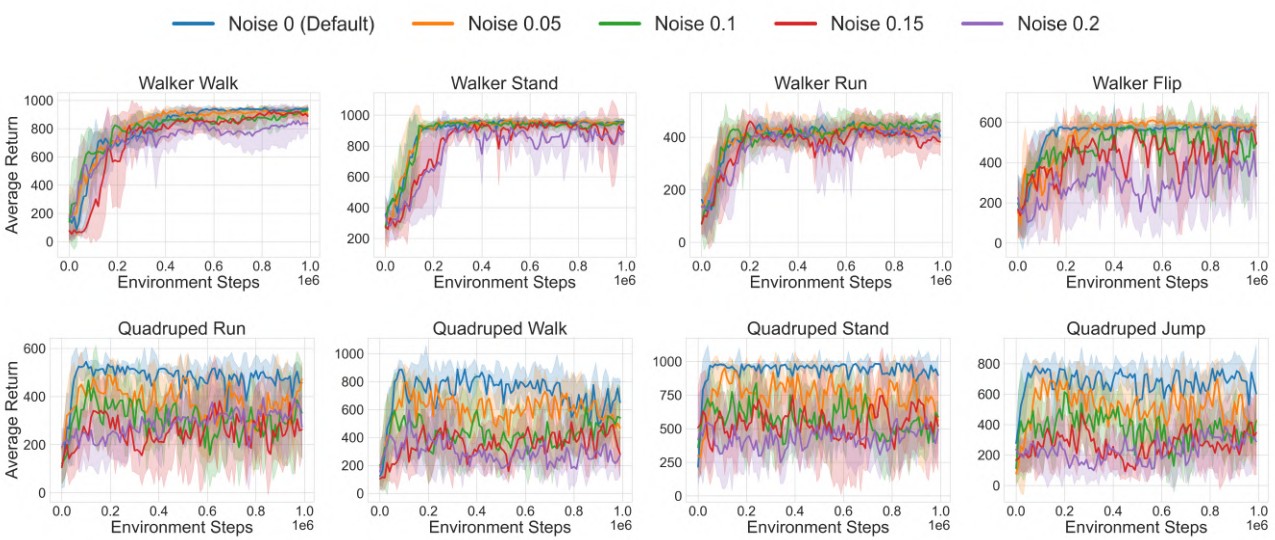

*Figure 14.* **Comparison between different noise settings.** In all plots, shaded regions indicate the standard deviation across 5 seeds.

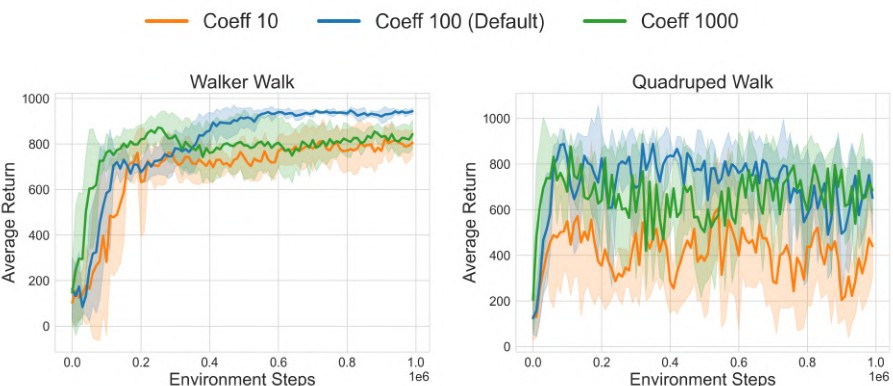

*Figure 15.* **Comparison between different contrastive coefficients.** In all plots, shaded regions indicate the standard deviation across 5 seeds.

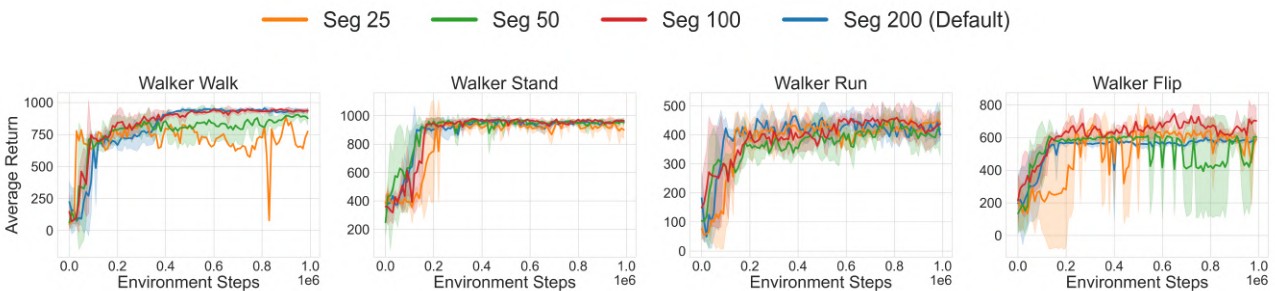

*Figure 16.* **Comparison between different segment sizes.** In all plots, shaded regions indicate the standard deviation across 3 seeds.

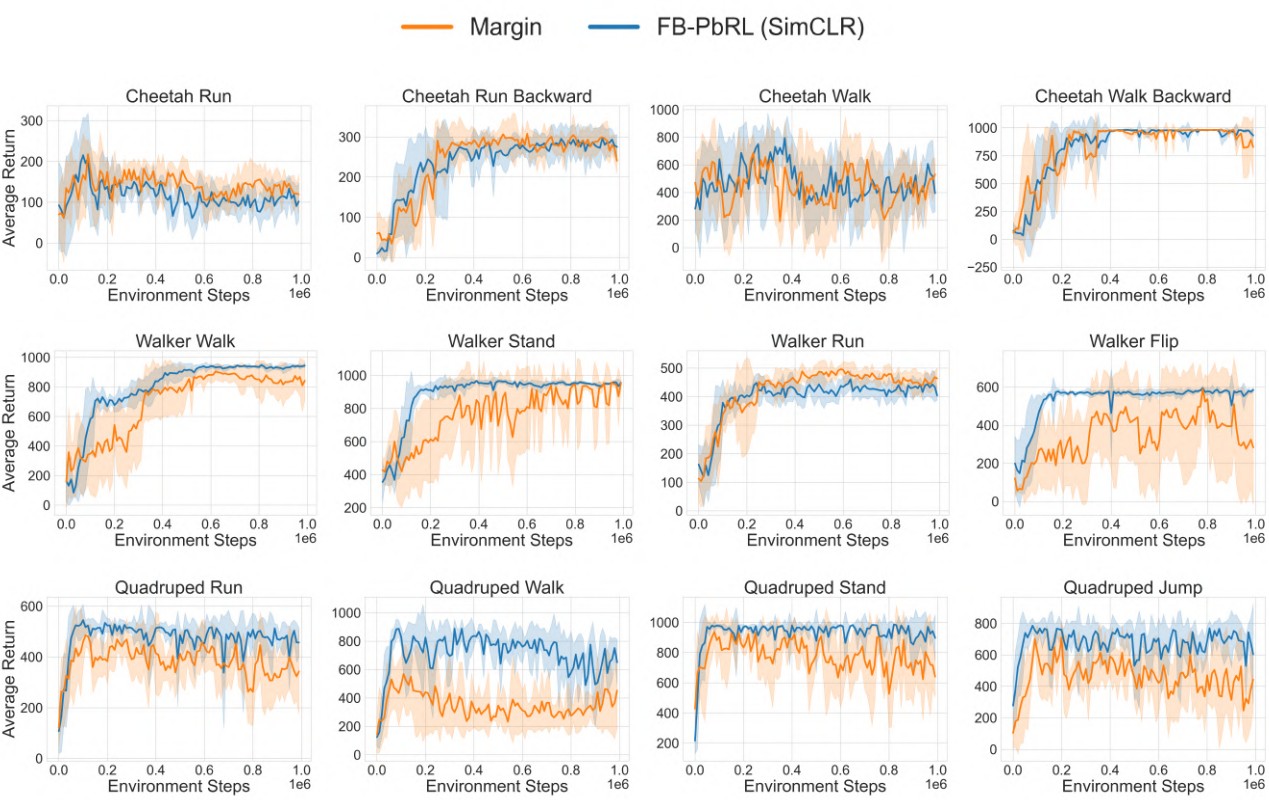

*Figure 17.* **Comparison between contrastive objectives.** In all plots, shaded regions indicate the standard deviation across 5 seeds.

