# OpenReview forum: "From Reward-Free Representations to Preferences: Rethinking Offline Preference-Based Reinforcement Learning"
_ICML.cc/2026/Conference — ICML 2026 regular_

### Official Review · Reviewer_sNTz · 2026-03-08

**Soundness:** 2
**Presentation:** 3
**Significance:** 3
**Originality:** 3
**Overall Recommendation:** 5
**Confidence:** 3

**Summary:**

This paper proposes a new framework for addressing the problem of preference-based reinforcement learning (PbRL) using reward-free representations, specifically, the forward-backward representation (FB). In PbRL, you are given an offline dataset of trajectories $\mathcal{D}$, and a preference dataset. As the FB only relies on a reward-free dataset, $\mathcal{D}$ can be used to learn the FB. To use the learned FB to compute a policy for a downstream reward, you usually need to use ground-truth downstream reward to fit a latent task specification vector $\mathbf{z}$. However, in the PbRL setting, you do not have access to ground-truth reward, and only has access to a preference dataset. The key contribution of this paper is to adapt the usual preference loss based on the Bradley-Terry model to directly learn this $\mathbf{z}$ using the preference dataset. The resulting method, coined FB-PbRL, sidesteps the need for learning a reward model, and benefits from the strong features learned using FB. Empirically, the paper demonstrates that such an approach achieves strong performance compared to state-of-the-art baselines.

**Compliance With Llm Reviewing Policy:**

Affirmed.

**Final Justification:**

I provided an initial rating of 4. While the paper is overall strong, I had some concerns, the most prominent ones related to (1) the difference of the performance of the proposed method in different tables, and (2) the decreased performance gain when the method is applied to real human preferences, both of which have been adequately addressed in the rebuttal. I have therefore raised my score to 5 to reflect this.

**Key Questions For Authors:**

1. In Table 3, where you compare performance in tasks with real human preferences, the performance gain of the proposed approach over DPPO is quite modest, which is different from the results in Table 2, where your approach significantly outperform DPPO when using synthetic preferences. Why is this the case? Could that suggest the benefits of the proposed approach compared to existing approaches are limited when using real human preferences, since, for example, real human data is scarce? Would this be another limitation of the proposed approach?
3. In Eq. 9, you consider only reward functions that lie in the span of the backward representation. Does that impose a restriction on the space of reward functions that can be considered?

**Limitations:**

see weakness 1, and key questions

---
I'm happy to raise my score if my concerns are addressed.

**Strengths And Weaknesses:**

Strengths:
- The paper is well-written and clear, with good figures to aid explanation of ideas.
- The idea of combining PbRL and FB, while conceptually simple, is novel and might influence future research in PbRL.
- Experiments are extensive, with comparison to a wide variety of baselines, demonstrating the strong performance of the proposed approach. Code is also provided for reproducibility.

Weaknesses:
1. Limitations of this work are not discussed. For example, how does the proposed method compare to existing PbRL methods in terms of computational complexity? It seems that to learn the FB, which is a general representation for all downstream rewards, you would need more compute than methods that solely focus on reward signals inferred from the preference dataset. Would increasing the compute used for existing PbRL methods close the performance gap in Table 1?
2. While the math is simple, it might still be better to include the detailed derivation of Eq. 13 in the Appendix to improve clarity. Also, shouldn't some constants (e.g., $k$ and $\tau$) appear in Eq. 13?
3. One concern regarding empirical results: Why is the performance of Ours-FT different in Tables 1 & 2? Aren't these numbers from the same method (Ours-FT) in the same domains?

---

> ### Author Rebuttal · Authors · 2026-03-31
>
> We thank the reviewer for the careful reading and insightful feedback.
> We address each concern point-by-point below.
>
> **Q1: Computational overhead compared to other PbRL methods**
>
> We thank the reviewer for raising this important point. Our method separates an amortized pre-training phase from task-specific fine-tuning. Since FB pre-training is performed once per domain and reused across tasks, its effective cost scales as $1/N_{\text{task}}$, making it favorable compared to end-to-end PbRL methods that retrain for every task.
> To illustrate, we report wall-clock time on the **Walker domain (4 tasks)** using RTX 3090 GPUs. The table below shows the time required for (i) FB pre-training / reward model training and (ii) fine-tuning / offline RL (1M steps) across the three strongest PbRL baselines.
>
> **Total Computation Time for 4 Walker tasks (hours)**
> |Phase|OPRL|CLARIFY|LiRE|FB-PbRL (Ours)|
> |:---| :--- | :--- | :--- | :--- |
> | FB Pretraining / Reward Model Training|24|24| - |18.5|
> | Finetuning / Offline RL|20|20|18|50|
>
> Notably, although the fine-tuning time of FB-PbRL appears larger (50h), this is because we report a fixed budget of **1M steps for all methods** to ensure fair comparison. In practice, FB-PbRL is substantially **more sample- and time-efficient**: it reaches competitive performance within ~200k steps and surpasses the strongest baseline (OPRL) in roughly **one hour of training**, while baselines plateau earlier.
>
> The training curves in terms of wall clock time are available at:
> https://imgur.com/a/04blxfY
>
> Therefore, despite a somewhat higher per-step wall-clock cost, FB-PbRL achieves better performance under equal wall-clock time budgets and converges significantly faster in practice.
>
> **Q2: Explain the theoretical derivation of SimCLR objective in FB-PbRL**
>
> Due to space constraints, we kindly refer the reviewer to our response to Reviewer 9E1p (Q1) for the detailed derivation of Eq. (13). To improve clarity, we will include the derivation in the appendix in the camera-ready version.
>
> **Q3: Why does Ours-FT perform differently across Tables 1 and 2?**
>
> Thank you for the careful reading and thoughtful question. We’d like to clarify that both results use Ours-FT but follow different evaluation protocols (Sec. 4.1).
> - In the PbRL Protocol (Table 1), we directly sample 2,000 preference pairs from the full offline dataset.
> - In contrast, the Zero-Shot RL Protocol (Table 2) first samples 400 trajectory segments (10,000 transitions) to match the supervision budget of zero-shot baselines, and then constructs preference pairs from this subset.
>
> This difference in data access and supervision leads to the observed performance gap.
>
> **Q4: Why smaller performance gain under real human preferences (Table 3)?**
>
> Thank you for the insightful question. We clarify that the smaller performance gap in Table 3 is primarily due to the high noise level in human preference labels, rather than data scarcity. We measure the noise ratio in the Adroit human preference datasets as the fraction of labels inconsistent with ground-truth rewards (excluding ties). We find that the noise ratio is extremely high: **60%** for pen-human and **55%** for pen-cloned. This indicates that human preferences are highly noisy relative to the synthetic preferences labeled from true rewards.
>
> To further validate this effect, we conduct experiments by increasing the noise ratio in synthetic preferences on quadruped tasks. As shown in the figure (https://imgur.com/a/2D0VLNZ), when the noise ratio reaches 50%, the performance gap between FB-PbRL and DPPO significantly shrinks, closely matching the trend in Table 3. This suggests that the reduced gap under real human preferences is mainly driven by high noise levels, rather than a limitation of our method.
>
>
> **Q5: Does Eq. (9) restrict the reward function space?**
>
> Thank you for the insightful question. In the FB literature (Touati & Ollivier, 2021; Theorem 14), it is shown that if a reward $r$ lies outside the span of backward representation $B$, the induced policy is optimal for the $L\_2$ orthogonal projection of $r$ onto $\text{span}(B)$.
>
> Specifically, for any bounded reward $r$ (not necessarily in $\text{span}(B)$) with optimal value $V^*\_r$, consider $\pi\_{z}$ with $z=\mathbb{E}[B(s,a)r\_{B}(s,a)]$ where $r\_B$ is the $L\_2$ orthogonal projection of $r$ onto $\text{span}(B)$. Then,
>
> $$ \lVert V^{\pi\_z} - V^*\_r \rVert\_\infty \leq \frac{2}{1-\gamma} \lVert r-r\_B \rVert\_{\infty}$$
>
> Thus, the method does not strictly restrict the reward space: performance degrades gracefully with the approximation error $\lVert r-r\_B \rVert\_{\infty}$, and $\pi\_z$​ remains near-optimal whenever $B$ captures the task-relevant components of $r$.

---

> > ### Author Rebuttal · Reviewer_sNTz · 2026-04-02
> >
> > Thank you for the detailed response. My concerns are sufficiently addressed. I have raised my score to 5 to reflect this.

---

> > > ### Author Response · Authors · 2026-04-07
> > >
> > > Thank you for acknowledging our clarifications and additional results. We sincerely appreciate your positive evaluation of our work. Your feedback has been invaluable in improving the paper.

---

### Official Review · Reviewer_RtxN · 2026-03-10

**Soundness:** 3
**Presentation:** 3
**Significance:** 2
**Originality:** 2
**Overall Recommendation:** 4
**Confidence:** 4

**Summary:**

This paper combines reward-free representation learning (RFRL) with preference-based RL (PbRL). The idea is to use a two-stage framework, 1) first learn successor-measure representations from unlabeled offline data, 2) then fine-tune using preference feedback. The motivation is reasonable and the setup is clear.

**Compliance With Llm Reviewing Policy:**

Affirmed.

**Ethical Review Concerns:**

None.

**Final Justification:**

My concerns were addressed appropriately.

**Key Questions For Authors:**

1. Can authors provide some theoretical analysis or guarantee showing when the learned representation is sufficient for optimal downstream control?
2. Can authors isolate whether the improvement mainly come from better representation quality, or better exploration that happened during pretraining? It is hard to tell from the current results.

**Limitations:**

There is no formal proof of representation sufficiency or optimality, which weaken the theoretical contribution. Also the ablations do not fully isolate each component, making it hard to understand what exactly driving the performance gain.

**Strengths And Weaknesses:**

- soundness
The motivation is well grounded, where reward-free environments is a valuable and practical problem. The proposed objective is clearly defined, and it can be seen how to connect to dynamics modeling. The experimental comparisons are also solid, including multiple baselines and ablations.
However, the theoretical justification feels a bit weak. The paper argues intuitively why the representation should capture task-relevant information, but there is no formal guarantee. It would be better to address this.
- presentation
The manuscript is logically organized and easy to follow. Figures and tables are both in good qualities.
But some notations are not well explained. For example, what is s' and a' in Section 2.2?
- significance
Reward-free RL is very active and important research direction. Learning reusable representation also has strong implication for sample efficiency and transfer.
One concern is this work only focuses on offline RL. But one very important PbRL work — the Christiano et al. 2017 NeurIPS paper which is also cited in introduction — is an online method. I think it will be important to compare with this, or at least discuss how the offline setting can extend to online setting. This is a bit limitation of the current work.
- originality
The paper combines dynamics-based intrinsic objectives with representation pretraining in a coherent way. The novelty is more in the formulation and integration side rather than some fundamentally new principle.

---

> ### Author Rebuttal · Authors · 2026-03-31
>
> We thank the reviewer for the careful reading and insightful suggestions.
>
> **Q1: Theoretical analysis of FB-PbRL**
>
> Thank you for the helpful suggestion. We can directly extend the FB analysis of (Touati & Ollivier, 2021) to PbRL, showing that the learned $F,B$ together with test-time search for $z$ (Eq. (13)) yield near-optimal control under standard conditions.
> - Let $F(s,a,z), B(s,a)$ be pre-trained representations such that for any reward $r$, $\pi_{z_r}$ is $\epsilon_{FB}$-optimal, i.e., $\lVert V^{\pi_{z_r}} - V^*_r \rVert\_{\infty}\le \epsilon\_{FB}$, where ${z}_r:=\mathbb{E}[r(s,a) B(s,a)]$, $\pi\_{z_r}=\arg\max\_{a}\ F(s,a,z_r)^\top z_r$, and $V^\ast_r$ is the optimal value for $r$.
> - Let $r_\psi$ be the true test-time reward (with optimal value $V^*$), $\hat{r}$ an estimate, and $\pi_{\hat{z}_r}=\arg\max\_{a}\ F(s,a,\hat{z}_r)^\top \hat{z}_r$.
> - For each $k$-step preference pair $(\sigma^+,\sigma^-)$, define $\phi(\sigma^+,\sigma^-):=\sum_{i=1}^{k}  B(s_i^+,a_i^+)-B(s_i^-,a_i^-)$ and $\Sigma_D:=\frac{1}{|D|}\sum_{(\sigma^+,\sigma^-)\in D} \phi \phi^\top \succeq 0$, with $\lambda_{\min}(\Sigma_D)$ its smallest eigenvalue.
>
> **Proposition**: For any $\lambda>0$, w.p. at least $1-\delta$,
>
> $$\lVert V^{\pi_{\hat{z}_r}}-V^*\rVert\_{\infty} = O\left(\frac{1}{(\lambda\_{\min}(\Sigma_D)+\lambda)}\sqrt{\frac{d\log (1/\delta)}{|D|}+\lambda}+\epsilon\_{FB}\right)$$.
>
> **Proof sketch**
>
> Step 1: By Proposition 10 in (Touati & Ollivier, 2021):
>
> $$\lVert V^{\hat{\pi}_{z_r}}-V^*\rVert\_\infty \le \frac{2 \lVert r\_{\psi} - \hat{r}\rVert\_\infty}{1-\gamma}+\epsilon\_{FB}$$
>
> Step 2: Bound $\lVert r_{\psi}-\hat{r}\rVert_\infty$. With $r_{\psi}=B(s,a)^\top\psi$ as Eq. (9) (wlog, let $\lVert\psi\rVert_2\le 1$), Lemma 5.1 in (Zhu et al., 2023) gives that MLE via preference loss in Eq. (2) can learn $\hat{\psi}$ that approximates the true $\psi$, i.e., for any $\lambda>0$, w.p. at least $1-\delta$,
>
> $$\lVert \psi-\hat{\psi}\rVert_{\Sigma_D+\lambda I}\le C_0\sqrt{\frac{d \log (1/\delta)}{|D|}+\lambda}$$
>
> where $C_0$ is some constant.
> Since Eq. (13) (SimCLR) is a reparameterization of Eq. (2), its solution satisfies $z_{\text{SimCLR}}=H_B \hat{\psi}$ (essentially finding $\hat{\psi}$). Thus,
>
> $$\lVert r_{\psi}-r_{\hat{\psi}}\rVert=\max_{(s,a)}\ | B(s,a)^\top(\psi-\hat{\psi})| \le \max_{(s,a)} \lVert B(s,a)\rVert_{(\Sigma_D +\lambda I)^{-1}}\cdot \lVert \psi-\hat{\psi} \rVert_{\Sigma_D+\lambda I}$$
>
> by Cauchy-Schwarz and $|a^\top b|=|(a^\top U^{-1/2})(U^{1/2}b)|$ ($U\succ 0$).
>
> Together, the bound shows that representation sufficiency reduces to preference data coverage $\lambda_{\min}(\Sigma_D)$ and estimation error, yielding near-optimal downstream control.
>
> **Q2: Can FB-PbRL extend to online PbRL settings?**
>
> Yes, FB-PbRL extends naturally to online PbRL by collecting trajectories with an online FB backbone (DVFB; Sun et al., 2025), followed by preference querying. As suggested, we compare online FB-PbRL with PrefPPO (Christiano et al., 2017) and observe consistent gains across all **quadruped** tasks.
>
> |Task|Online FB-PbRL|PrefPPO|
> |-|-|-|
> |walk|**597.2 ± 60.0**|279.7 ± 8.7|
> |run|**425.9 ± 77.8**|198.6 ± 26.3|
> |stand|**958.5 ± 1.2**|357.4 ± 4.6|
> |jump|**679.2 ± 41.5**|298.9 ± 21.3|
> |Average|**665.2**|283.7|
>
> These show that FB-PbRL extends effectively beyond the offline setting and outperforms a standard online PbRL baseline.
>
> **Q3: Disentangle if the gains come from better representations or better exploration?**
>
> Based on the ablations (Sec 4.3), we can explicitly disentangle representation quality and alignment of $z$ in two steps.
> - Comparing OPPO and FB-BT-FT, both use learned representations but differ in how they are obtained. FB-BT-FT consistently outperforms OPPO, showing that FB pre-training yields higher-quality representations.
> - Comparing FB-BT-FT with full FB-PbRL isolates the benefit of reward-aligned latent $z$ (Fig. 3(b)): FB-PbRL further improves performance, showing that the preference-guided fine-tuning via the contrastive objective better aligns representations to identify optimal task vector $z$.
> |Domain|OPPO|FB-BT-FT|FB-PbRL (Ours)|
> |-|-|-|-|
> |Cheetah|258.4±26.4| 536.6±42.4|**621.7±16.1**|
> |Walker|219.6±3.7|600.6±34.9|**794.5±31.2**|
> |Quadruped |398.7±33.3|714.1±17.5|**846.9±10.7**|
>
> Finally, since all methods use the same offline dataset (w/o online exploration), the improvements do not arise from differences in data coverage but from improved representation and alignment.
> (If our interpretation of exploration differs from your intent, we would sincerely appreciate any further clarification. Thanks again for the thoughtful feedback)
>
> **Q4: Notation $s’,a’$ in Sec 2.2**
>
> $s’$ and $a’$ denote the next state and next action, following standard RL notation.
>
> ---
> Zhu et al., "Principled reinforcement learning with human feedback from pairwise or k-wise comparisons," ICML 2023.
>
> Sun et al., "Unsupervised zero-shot reinforcement learning via dual-value forward-backward representation," ICLR 2025.

---

> > ### Author Rebuttal · Reviewer_RtxN · 2026-04-03
> >
> > My concerns were addressed. I have raised my score.

---

> > > ### Author Response · Authors · 2026-04-07
> > >
> > > Thank you for your responses and for the time and effort devoted to reviewing our paper and providing thoughtful suggestions. We are pleased that our clarifications have addressed your concerns, and we will do our best to incorporate these into the final version.

---

### Official Review · Reviewer_NZe7 · 2026-03-11

**Soundness:** 2
**Presentation:** 3
**Significance:** 2
**Originality:** 3
**Overall Recommendation:** 4
**Confidence:** 2

**Summary:**

This paper proposes FB-PbRL, which connects RFRL with offline PbRL. It leverages the Forward-Backward decomposition technique from RFRL to learn latent state-action representations from reward-free offline data, then apply a contrastive objective over preference-labeled trajectories to jointly fine-tune representations and search for a latent task vector. This eliminates the need to explicitly learn a reward or preference model.

**Compliance With Llm Reviewing Policy:**

Affirmed.

**Final Justification:**

The authors’ rebuttal addressed my main concerns regarding the contrastive reformulation and the experimental evaluation.

**Key Questions For Authors:**

- Eq. (11), is the matrix H_B guaranteed invertible in practice, and how is singularity handled during optimization?

- PointMass Bottom Right in Table 1 and the PointMass average in Table 2 appear weaker or inconsistent relative to other domains. Why would this happen?

**Limitations:**

yes

**Strengths And Weaknesses:**

**Strengths**:

- This work connects RFRL to PbRL, leveraging latent successor measures, and Reformulates preference loss as a contrastive objective akin to SimCLR.

- This work shows strong empirical performance FB-PbRL, which consistently outperforms all offline PbRL baselines.

- It is robustness to limited and noisy preferences.


**Weaknesses**:

- Contrastive reformulation: "L_pref​ is convex in z"" is made for Eq. (13), but the actual optimized objective in Eq. (18) uses cosine similarity and is updated jointly with B_ω​, for which convexity is not established. This matters because the optimization advantage is part of the method’s motivation.

- Experiments: 1) Unfair comparison with baselines. FB-PbRL benefits from RFRL pre-training on 5 million reward-free transitions per domain, whereas PbRL baselines have access only to the same preference dataset and the offline dataset for downstream RL. 2) Insufficient analysis. The failure case in "Bottom Right" task is acknowledged only in passing and lacks analysis. The high variance are observed in certain tasks. 3) A direct comparison of total compute budget (pretrain + finetune) across methods and ablation in loss coefficients λ and α are absent.

- typos: miss a closing parenthesis in line 251 (denoted by z_PGFT. The notation z_CPTS and z_PGFT are introduced in Sec. 3 but Algorithm 1 uses only z without subscript. "a detailed version in Appendix 2." -> Algorithm 2.

---

> ### Author Rebuttal · Authors · 2026-03-31
>
> We thank the reviewer for the thoughtful and constructive feedback. We address each concern point-by-point below.
>
> **Q1: Fairness in comparison – Is FB-PbRL using more data than PbRL baselines?**
>
> We clarify that the 5M reward-free transitions used by FB-PbRL are from exactly the same offline dataset available to all PbRL baselines. The difference is only in usage: FB-PbRL leverages this dataset for reward-free pretraining, while baselines use it solely for downstream offline RL.
> Importantly, FB-PbRL does **not** access any additional data during fine-tuning. Thus, the comparison is fair in terms of data, differing only in how the shared offline dataset is utilized.
>
> **Q2: Ablation in loss coefficients $\lambda$ and $\alpha$**
>
> For $\lambda$, we set $\lambda=1$ following the original FB pre-training setup (Touati et al., 2023). To validate this choice, we additionally evaluate $\lambda\in\{0.3, 3\}$ on walker-walk and quadruped-walk. The results show that $\lambda=1$ consistently achieves the best performance, confirming prior findings.
>
> |$\lambda$|0.3|**1**|3|
> |-|-|-|-|
> |Walker-walk|898.6|**961.5**|890.6|
> |Quadruped-walk|932.0|**944.2**|925.5|
>
> For $\alpha$, we already include an ablation study in Fig. 5(c) in the original manuscript. To further strengthen this analysis, we additionally evaluate the **Walker** tasks as below:
>
> |$\alpha$|10|**100**|1000|
> |-|-|-|-|
> |walk|888.0|**961.5**|923.9|
> |stand|970.9|**980.3**|978.2|
> |run|491.5|**503.7**|469.3|
> |flip|584.1|**606.0**|578.7|
>
> Our method demonstrates stable performance across a wide range of values, indicating it is not sensitive to this hyperparameter. We use $\alpha=100$ as the default since it consistently yields the strongest results.
>
> **Q3: Why weaker performance in "Bottom Right" in Table 1 and higher variance and lower average score in some PointMass tasks in Table 2?**
>
> Thank you for the suggestion. The weaker performance on PointMass (especially Bottom Right) primarily arises from severe dataset imbalance in the RND PointMass Maze. As shown in visitation heatmap (https://imgur.com/a/welKBcJ), samples are concentrated in the top-left region, while the bottom-right is significantly underrepresented. This becomes even more evident when focusing on low-visit states (https://imgur.com/a/hLTybV7), leading to insufficient supervision in that region.
>
> Moreover, our Zero-Shot RL evaluation protocol (Sec 4.1) builds the preference dataset from only 10k transitions. To ensure coverage, we adopt a short segment length of 25 following (Choi et al., 2024), which yields shorter and less informative trajectories and hence a challenging scenario. As shown in trajectory visualizations (https://imgur.com/a/0skXGqZ), this results in short and fragmented trajectories, offering weaker preference signals and hence leads to higher variance in the achieved score in Table 2.
>
> **Q4: Comparison of total compute budget to PbRL methods**
>
> Due to space constraints, we kindly refer the reviewer to our response to Reviewer 9E1p (Q2) for details.
>
> **Q5: Convexity of Eq. (13) and Eq. (18)**
>
> We agree that convexity should be stated precisely. In Eq. (18), although cosine similarity is used, the optimization is still effectively performed over normalized latent vectors. Since $z$ and $z\_\sigma$ are normalized by default in the FB framework, the cosine similarity reduces to an inner product, i.e., $\cos(z,z\_\sigma)=z^\top z\_\sigma$. Therefore, the objective retains the same structure as Eq. (13), and remains convex with respect to $z$.
>
> On the other hand, we agree that when jointly optimizing the backward representation through a neural network, the overall objective is no longer convex. Importantly, our convexity statement corresponds to the test-time search for $z$, where the representation is frozen and we solve for the optimal latent $z$. In this setting, the convexity property still holds and provides the intended optimization advantage.
>
> **Q6: Is $H_B$ guaranteed invertible in practice, and how is singularity handled?**
>
> In our framework, the invertibility of $H\_B$ is maintained inherently by the FB objective, eliminating the need for singularity handling. The auxiliary orthonormality loss regularizes $H\_B$ to approximate the identity matrix ($I\_d$). This actively prevents feature collapse and pushes $H\_B$ away from singularity. We empirically validated this stability following the feature rank protocol introduced by Touati et al., 2023. By evaluating $H\_B$ during fine-tuning across all four walker tasks, we confirm all eigenvalues remained strictly above the $10^{-4}$ threshold. Thus, $H\_B$ successfully maintains full rank and strict invertibility in practice.
>
> **Q7: Typos & notation inconsistencies**
>
> Thank you for catching these. We will fix them in the final version.
>
> ---
>
> Choi et al., "Listwise reward estimation for offline preference-based reinforcement learning," ICML 2024.
>
> Touati et al., “Does zero-shot reinforcement learning exist?,” ICLR 2023.

---

> > ### Author Rebuttal · Reviewer_NZe7 · 2026-04-02
> >
> > Thank you for the detailed rebuttal. The authors have addressed my main concerns reasonably well.  I will keep my score.

---

> > > ### Author Response · Authors · 2026-04-07
> > >
> > > We sincerely appreciate your recognition of our clarifications and added results. Thank you for your positive assessment of our work. Your comments have been instrumental in strengthening the paper.

---

### Official Review · Reviewer_9E1p · 2026-03-27

**Soundness:** 3
**Presentation:** 3
**Significance:** 3
**Originality:** 3
**Overall Recommendation:** 4
**Confidence:** 4

**Summary:**

This paper introduces FB-PbRL, a novel two-stage framework for offline Preference-based Reinforcement Learning (PbRL) that leverages Reward-Free Representation Learning (RFRL). The authors establish a theoretical connection between the standard preference loss in PbRL and SimCLR-style contrastive learning when applied to Forward-Backward (FB) representations. The proposed method first pre-trains FB representations using reward-free offline data. In the second stage, it employs a contrastive objective over preference-labeled trajectory segments to jointly fine-tune these representations and search for a latent task vector, eliminating the need to learn an explicit reward or preference model. Empirical evaluations on DeepMind Control Suite, Adroit, and MetaWorld benchmarks demonstrate that FB-PbRL achieves superior or competitive performance compared to state-of-the-art PbRL and zero-shot RFRL baselines, particularly in data-scarce and noisy preference settings.

**Compliance With Llm Reviewing Policy:**

Affirmed.

**Final Justification:**

I keep my score.

**Key Questions For Authors:**

See Weakness

**Strengths And Weaknesses:**

Strengths

1. The paper provides a compelling and underexplored theoretical link between PbRL and RFRL, reinterpreting preference learning through the lens of contrastive representation learning. This offers a fresh perspective on how to efficiently leverage human feedback.

2. The proposed FB-PbRL framework effectively bypasses the common pitfalls of reward-model over-optimization by directly aligning latent representations with preferences via a contrastive loss. The two-stage design (pre-training + preference-guided fine-tuning) is well-motivated and clearly presented.

3. The experimental evaluation is extensive and robust. FB-PbRL consistently outperforms a wide range of baselines (DPPO, OPPO, CLARIFY, LiRE, FB, HILP, etc.) across diverse domains (locomotion, navigation, manipulation). The method demonstrates impressive robustness to limited data and noisy feedback, which is critical for practical applications.

Weaknesses

1. While the connection to SimCLR is a key insight, the theoretical derivation is somewhat informal. The claim that the preference loss can be "equivalently reformulated as a contrastive objective" is stated under "mild structural conditions," but these conditions are not rigorously defined or proven within the main text. A more formal proof or a detailed discussion of these conditions would significantly strengthen the paper.

2. The two-stage process, particularly the RFRL pre-training, is computationally intensive. The paper notes that pre-training took ~20 hours and fine-tuning ~10 hours. While the method is effective, this cost could be a barrier for practitioners. A more detailed analysis of the computational overhead compared to end-to-end PbRL methods (e.g., DPPO) would be beneficial.

3. The derivation linking the preference loss to the SimCLR objective appears to rely on the specific structure of the FB representations. It is not immediately clear if this connection holds for other RFRL methods (e.g., Laplacian, HILP, PSM). Discussing the generalizability of this insight beyond the FB framework would add depth.

---

> ### Author Rebuttal · Authors · 2026-03-31
>
> We thank the reviewer for the thoughtful and constructive feedback. We address each concern point-by-point below.
>
> **Q1: Explain the theoretical derivation of SimCLR objective in FB-PbRL**
>
> Thank you for the helpful suggestion. We agree the description of Eq. (13) was a bit too concise and clarify both the steps and required conditions below.
>
> **(1) Preference loss derivation**
>
> Starting from Eq. (2), the preference loss is
>
> $$L\_{\mathrm{pref}}(\psi) = -\mathbb{E}\_{(\sigma^+,\sigma^-)\sim \mathcal{D}\_{\mathrm{pref}}} \left[ \log \mu\left(\frac{R\_\psi(\sigma^+) - R\_\psi(\sigma^-)}{\tau}\right)\right]$$
>
> ($\mu(\cdot)$: sigmoid function; $R\_\psi(\cdot)$: trajectory-level reward)
>
> By the reward parameterization in Eq. (9) and definitions in Eqs. (12) & (14),
>
> $$L\_{\mathrm{pref}}(\psi) = -\mathbb{E} \left[ \log \mu \left( \frac{k}{\tau} (z\_{\sigma}^+-z\_{\sigma}^-)^\top \psi\right) \right]$$
>
> where $k$ emerges since $z\_\sigma:= \frac{1}{k}\sum\_{t=1}^k B\_{\bar{\omega}}(s\_t,a\_t)$ is defined as the average backward embedding over a $k$-step segment.
>
> Substituting $\psi=H\_B^{-1} z$ (Eq. (11)),
>
> $$L\_{\mathrm{pref}}(z) = -\mathbb{E} \left[ \log \frac{ \exp(\frac{k}{\tau} z^\top H\_B^{-1} z\_{\sigma}^+) }{ \exp(\frac{k}{\tau} z^\top H\_B^{-1} z\_{\sigma}^+) + \exp(\frac{k}{\tau} z^\top H\_B^{-1} z\_{\sigma}^-) } \right]$$
>
> Here we set the temperature $\tau = k$. Thus, the segment length $k$ acts as a temperature parameter and is absorbed into the logits, yielding Eq. (13).
>
> **(2) Formalizing the "mild structural conditions”**
>
> The equivalence to a SimCLR objective holds under the conditions below
> - Linear reward parameterization: $r_{\psi}=B(s,a)^\top \psi$
> - Orthonormality of backward embedding: $H\_B = I\_d$.
>
> Under this, the loss simplifies to:
>
> $$L\_{\mathrm{pref}}(z) = -\mathbb{E} \left[ \log \frac{ \exp(z^\top z\_{\sigma}^+) }{ \exp(z^\top z\_{\sigma}^+) + \exp(z^\top z\_{\sigma}^-)} \right]$$
>
> These conditions are standard in FB and make explicit the equivalence to SimCLR.
>
> **Q2: Computational overhead compared to other PbRL methods**
>
> We thank the reviewer for raising this important point. Our method separates an amortized pre-training phase from task-specific fine-tuning. Since FB pre-training is performed once per domain and reused across tasks, its effective cost scales as $1/N\_{\text{task}}$, making it favorable compared to end-to-end PbRL methods that retrain for every task.
> To illustrate, we report wall-clock time on the **Walker domain (4 tasks)** using RTX 3090 GPUs. The table below shows the time required for (i) FB pre-training / reward model training and (ii) fine-tuning / offline RL (1M steps) across the three strongest PbRL baselines.
>
> **Total Computation Time for 4 Walker tasks (hours)**
> |Phase|OPRL|CLARIFY|LiRE|FB-PbRL (Ours)|
> |:---| :--- | :--- | :--- | :--- |
> | FB Pretraining / Reward Model Training|24|24| - |18.5|
> | Finetuning / Offline RL|20|20|18|50|
>
> Notably, although the fine-tuning time of FB-PbRL appears larger (50h), this is because we report a fixed budget of **1M steps for all methods** to ensure fair comparison. In practice, FB-PbRL is substantially **more sample- and time-efficient**: it reaches competitive performance within ~200k steps and surpasses the strongest baseline (OPRL) in roughly **one hour of training**, while baselines plateau earlier.
>
> The training curves in terms of wall clock time are available at:
> https://imgur.com/a/04blxfY
>
> Therefore, despite a somewhat higher per-step wall-clock cost, FB-PbRL achieves better performance under equal wall-clock time budgets and converges significantly faster in practice.
>
> **Q3: Does SimCLR connection generalize beyond FB?**
>
> The SimCLR connection in FB-PbRL (Eq. (13)) is not specific to FB, but applies more broadly to RFRL methods that parameterize rewards via a latent vector, e.g., HILP (Park et al., 2024).
>
> In HILP, Hilbert representations $\phi(s)$ are pre-trained and used to define rewards $r\_z(s,s')=\tilde{\phi}(s,s')^\top z$, where $\tilde{\phi}(s,s'):=\phi(s')-\phi(s)$. A latent-conditioned policy $\pi(a|s,z)$ is learned to achieve optimality for $r\_z$​, and test-time control reduces to searching for $z$ that matches the target reward.
>
> In the PbRL setting, plugging $r_z(s,s')$ into the preference loss (Eq. (2)) and following the same derivation as Eq. (13) (please see our response to Q1) yields a SimCLR-type objective:
>
> $$L\_{\mathrm{pref,HILP}}(z) = -\mathbb{E} \left[ \log \left(\frac{ \exp(z^\top \tilde{\phi}^{+}) }{\exp(z^\top \tilde{\phi}^{+}) +  \exp(z^\top \tilde{\phi}^{-})}\right) \right]$$
>
> where $\tilde{\phi}^{+}:=(\sum\_{t=1}^{k} \tilde{\phi}(s\_t^+, s\_{t+1}^+))/k$ and $\tilde{\phi}^{-}:=(\sum\_{t=1}^{k} \tilde{\phi}(s\_t^-, s\_{t+1}^-))/k$.
>
> This shows the connection arises from the latent reward structure, rather than FB-specific properties, and thus extends to other RFRL methods with similar parameterizations.
>
> ---
> Park et al., “Foundation policies with Hilbert representations,” ICML 2024.

---

> > ### Author Rebuttal · Reviewer_9E1p · 2026-04-03
> >
> > My concerns have been adequately addressed.

---

> > > ### Author Response · Authors · 2026-04-07
> > >
> > > Thank you for your responses and for the time and effort you invested in reviewing our paper and offering thoughtful suggestions. We are glad our clarifications have resolved your concerns and will strive to incorporate these results in the final version.

---

### Decision · Program_Chairs · 2026-04-30

**Decision:**

Accept (regular)

**Comment:**

The paper proposes FB-PbRL, bridging reward-free representation learning and offline preference-based RL by pretraining FB representations on reward-free offline data and then using a SimCLR-style contrastive objective on preference pairs to infer a latent task vector without learning an explicit reward model.

Reviewers find the idea novel, clearly presented, and empirically strong across DMControl/Adroit/MetaWorld, with good robustness to limited/noisy preferences.

Main concerns were about the conditions of the contrastive equivalence and convexity claims, compute overhead, and a few inconsistent results (e.g., PointMass performance, smaller gains with real human preferences). The rebuttal addresses theses concerns.

Overall, I recommend accept, with a request to add a clear derivation/assumptions and a brief limitations/compute discussion in the final version.